# Reprogramming of human peripheral blood mononuclear cells into induced mesenchymal stromal cells using non-integrating vectors

Wanqiu Chen[1], Chenguang Wang[1,2], Zhi-Xue Yang[3,4], Feng Zhang[3,4], Wei Wen[3,4], Christoph Schaniel [5], Xianqiang Mi[2], Matthew Bock [6], Xiao-Bing Zhang [3✉], Hongyu Qiu [7✉] & Charles Wang [1,8✉]

Mesenchymal stromal cells (MSCs) have great value in cell therapies. The MSC therapies have many challenges due to its inconsistent potency and limited quantity. Here, we report a strategy to generate induced MSCs (iMSCs) by directly reprogramming human peripheral blood mononuclear cells (PBMCs) with OCT4, SOX9, MYC, KLF4, and BCL-XL using a nonintegrating episomal vector system. While OCT4 was not required to reprogram PBMCs into iMSCs, omission of OCT4 significantly impaired iMSC functionality. The omission of OCT4 resulted in significantly downregulating MSC lineage specific and mesoderm-regulating genes, including *SRPX*, *COL5A1*, *SOX4*, *SALL4*, *TWIST1*. When reprogramming PBMCs in the absence of OCT4, 67 genes were significantly hypermethylated with reduced transcriptional expression. These data indicate that transient expression of OCT4 may serve as a universal reprogramming factor by increasing chromatin accessibility and promoting demethylation. Our findings represent an approach to produce functional MSCs, and aid in identifying putative function associated MSC markers.

[1] Center for Genomics, School of Medicine, Loma Linda University, Loma Linda, CA, USA. [2] Shanghai Institute of Microsystem and Information Technology Chinese Academy of Sciences, Shanghai, China. [3] Department of Medicine, Loma Linda University, Loma Linda, CA, USA. [4] State Key Laboratory of Experimental Hematology, Tianjin, China. [5] Division of Hematology and Medical Oncology, Black Family Stem Cell Institute, Tisch Cancer Institute, Mount Sinai Institute for Systems Biomedicine, Icahn School of Medicine at Mount Sinai, New York, NY, USA. [6] Department of Pediatrics, School of Medicine, Loma Linda University, Loma Linda, CA, USA. [7] Translational Cardiovascular Research Center, Department of Internal Medicine, University of Arizona - College of Medicine at Phoenix, Phoenix, AZ, USA. [8] Division of Microbiology & Molecular Genetics, Department of Basic Sciences, School of Medicine, Loma Linda University, Loma Linda, CA, USA. ✉email: zhangxbhk@gmail.com; hqiu@arizona.edu; chwang@llu.edu

Mesenchymal stromal cells (MSCs), also termed mesenchymal stem cells in the literature, have long been proposed for use in regenerative medicine and cell therapy. Currently, their therapeutic effects are being tested in more than 1300 clinical trials worldwide, involving a wide variety of diseases (www.clinicaltrials.gov). However, there are still many challenges and controversies regarding MSC-based clinical applications due to their cellular heterogeneity, inconsistent potency, limited quantity, and rapid cellular senescence during ex vivo expansion[1]. Therefore, establishing an approach to generate large amounts of MSCs, especially autologous MSCs, with consistent potency, which can be quality controlled using reliable molecular identifiers, will significantly facilitate the clinical use of MSCs.

Peripheral blood cells have been widely used for reprogramming into induced pluripotent stem cells (iPSCs)[2,3], since phlebotomy is minimally invasive and blood is the most readily accessible cell source in the human body. Compared to fibroblasts, blood cells are less likely to acquire genetic mutations induced by environmental insults[4,5], thus providing them with a potential safety advantage. Retroviral or lentiviral vectors for reprogramming may induce insertional mutagenesis and thus raise safety concerns. Episomal vectors are nonintegrating plasmids that have been used to reprogram fibroblasts or blood cells into transgene- and virus-free iPSCs[3,6]. The episomal plasmid within the reprogrammed cells will not be detected after ~5 passages[7,8]. To increase the expression of reprogramming factors in blood cells, we modified the episomal vectors to include the strong spleen focus-forming virus (SFFV) promoter and the Woodchuck hepatitis virus posttranscriptional regulatory element (WPRE), which led to a 10- to 100-fold[9,10] improvement in reprogramming efficiency.

Previously, we reported that fetal cord blood CD34$^+$ cells can be directly reprogrammed into induced MSCs (iMSCs) by lentiviral delivery of OCT4 alone[11]. In this study, we chose adult peripheral blood mononuclear cells (PBMCs) as source cells due to their easy accessibility and availability compared to cord blood. We aim to establish a simple system to easily generate large amounts of clinically relevant, integration-free iMSCs from a few milliliters of peripheral blood. In the present study, we found that unlike fetal CD34$^+$ cells, a single factor OCT4 failed to reprogram adult PBMCs into iMSCs; however, through transient overexpression of five factors by episomal vectors (OCT4, BCL-XL, MYC, KLF4, and SOX9), adult PBMCs can be highly efficiently reprogrammed into integration-free iMSCs with trilineage differentiation potential.

Epigenetic constraints are the most formidable barrier to efficient cellular reprogramming. Dynamic changes in the transcriptome and epigenome, including DNA methylation and chromatin openness, occur during the reprogramming process[12]. The key mechanistic question of reprogramming to the MSC state is how the parental cells' epigenetic signature is erased, and a MSC signature is established. Epigenetic regulation is essential for proper control of the maintenance of MSC self-renewal versus differentiation[13]. Considering the elusive surface marker for MSCs, the epigenetic signature has been proposed as a quality check to better characterize MSCs[14]. In this study, we obtained DNA methylome by reduced representation of bisulfite sequencing (RRBS), global chromatin accessibility by ATAC-seq, and transcriptome by RNA-seq to understand the mechanism of iMSC reprogramming. The gained insight facilitates a deeper understanding of differential iMSC functions. The unveiled epigenetic signature of functional iMSCs will aid in developing strategies for efficient iMSC reprogramming and establishing molecular standards for characterizing functional MSCs with therapeutic potential.

## Results

**OCT4 alone was insufficient to reprogram PBMCs into iMSCs directly.** Previously, we reported that lentivirally expressed OCT4 could directly reprogram human cord blood CD34$^+$ hematopoietic progenitor cells into iMSCs with very high efficiency[11]. Therefore, we first tried to convert human PBMCs into iMSCs by overexpressing OCT4 alone using a clinically relevant vector system. Isolated human PBMCs were cultured in a Stemline-based erythroid medium for six days to expand erythroid progenitors. Using the nucleofection method, $2 \times 10^6$ expanded PBMCs were transfected with our modified oriP/EBNA1-based episomal vector, which expressed OCT4 under a strong SFFV promoter (Fig. 1a), as we previously described[11]. Cells were then cultured in MSC medium[11] supplemented with small molecules that promote reprogramming (3 μM CHIR99021, 10 μM forskolin, 10 μM ALK inhibitor (SB431542), and 5 μM tranylcypromine hydrochloride)[15]. However, there was no MSC-like colony formation 2 weeks later, indicating that OCT4 alone was insufficient to convert human PBMCs into iMSCs directly (Fig. 1b).

**Generation of iMSCs from human PBMCs using five factors.** Our previous studies showed that BCL-XL is a critical reprogramming factor in blood cell reprogramming[9,16], which increased the reprogramming efficiency by 10-fold when converting PBMCs into iPSCs using Yamanaka factors[16]. Here, we observed that transfection of PBMCs with OCT4, BCL-XL, and MYC (OBM) led to the formation of MSC-like colonies 2 weeks later (Fig. 1b), albeit at low efficiency. The combination of any two of the OBM factors failed to generate iMSC colonies (Fig. 1b). To improve the reprogramming efficiency further, we examined OBM with different combinations of other factors for generating iPSCs, including KLF4 and SOX2. KLF4 moderately improved iMSC generation, whereas SOX2 increased reprogramming efficiency by ~5-fold (Fig. 1c and Supplementary Data 1). However, the presence of SOX2 in the reprogramming cocktail resulted in ~1–2% of reprogrammed cells expressing iPSC markers, e.g., TRA-1-60 (Fig. 1d) and NANOG (Supplementary Fig. 1), even in MSC expansion culture conditions. Since iPSCs may induce teratomas, the SOX2-containing approach is not clinically prudent.

We decided to replace SOX2 with SOX9 because SOX9 plays an important role in skeletal development and chondrogenesis[17,18]. Surprisingly, SOX9 showed greater potency than SOX2 in iMSC reprogramming (Fig. 1c). As expected, SOX9 virtually abolished the generation of TRA-1-60-expressing cells (Fig. 1d). To ensure the absence of undetectable levels of iPSCs after reprogramming with SOX9, we cultured iMSCs in iPSC medium for 1 week. Phenotyping analysis of cultured cells showed no expression of iPSC markers. These data suggested that SOX9 restricted cell fate to iMSCs, whereas SOX2 would overshoot the reprogramming of a proportion of PBMCs beyond the stage of iMSCs. Moreover, after reprogramming with SOX9, PBMCs transformed morphologically to spindle-like cells resembling MSCs within 4–6 days, whereas SOX2-reprogrammed cells did not display spindle-like morphology (Supplementary Fig. 2a).

Although PBMCs are composed of many different cell types, based on our previous studies[3,16,19], we hypothesized that the CD34$^+$ cell subset in peripheral blood was the most amenable to reprogramming to iMSCs. After six days of culture in hematopoietic stem cell expansion medium, the percentage of CD34$^+$ cells in PBMCs increased from <1% to ~4–5%. When we depleted CD34$^+$ cells from PBMCs before inducing reprogramming, no MSC-like colonies were observed (Fig. 1e). These results suggested that the five reprogramming factors converted the CD34$^+$ hematopoietic stem cells and progenitors but not the mature blood cells into iMSCs.

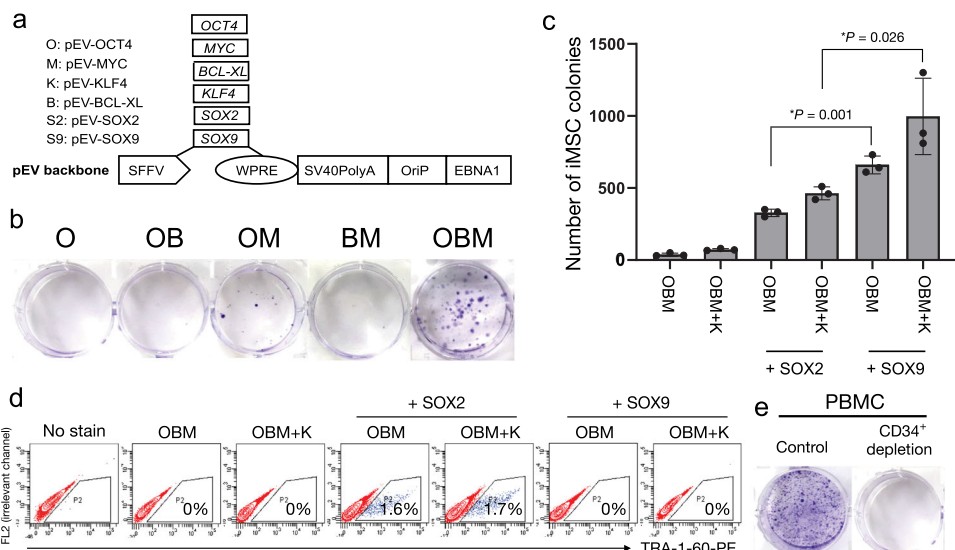

**Fig. 1 OCT4 alone was insufficient to direct reprogramming of human PBMCs into iMSCs. a** Schematic diagram of the episomal vector plasmids. SFFV is the spleen focus-forming virus U3 promoter; WPRE, posttranscriptional regulatory element; SV40PolyA, polyadenylation signal from SV40 virus; OriP, EBV (Epstein–Barr virus) origin of replication; EBNA1, Epstein–Barr nuclear antigen 1. **b** Colony formation at day 14 after nucleofection with $2 \times 10^6$ PBMCs and maintenance in MSC culture conditions. **c** Reprogramming efficiency with different combinations of reprogramming factors. Error bars indicate standard deviation. $n = 3$ biologically independent samples for each group. **d** Fluorescence-activated cell sorting (FACS) analysis of iMSCs 8 days after reprogramming with different factor combinations. SOX2 induced iPSCs generation (TRA-1-60+ cells). However, SOX9 did not induce detectable TRA-1-60+ cells. **e** Colony formation at day 14 after nucleofection with $1 \times 10^6$ PBMCs (control) or CD34+-depleted PBMCs followed by maintenance in MSC culture conditions.

Having observed that the combination of OCT4, BCL-XL, MYC, KLF4, and SOX9 (named as 5F) induced the highest levels of PBMC conversion without overshooting the iMSC reprogramming process, we used the five factors (5F) for reprogramming in subsequent experiments. In all, 5–7 days after nucleofection of PBMCs with 5F, dozens of MSC-like colonies were observed. At approximately 2 weeks, reprogrammed cells resembled MSCs with typical spindle-like morphology (Fig. 2a). The expression of MSC markers such as CD90 and CD73 increased from ~5% of reprogrammed cells by ~1 week to ~15% and 40% of the cells, respectively, by week 2 and >75% by week 3 (Fig. 2b and Supplementary Data 2). Four weeks after reprogramming with 5F, almost all cells expressed typical MSC markers: CD29 (99.7%), CD73 (95.3%), CD90 (96%), and CD166 (80%) (Fig. 2c, d). The expression of hematopoietic markers such as CD45 and CD34 was negligible (Fig. 2e). In addition, OCT4+ cells were not detectable (Supplementary Fig. 3). Next, we evaluated the immunomodulatory potential of the iMSCs. We found that our 5F iMSCs were able to significantly suppress T-cell proliferation (CD4+ and CD8+ T-cell subsets) after 3 or 6 days (Fig. 2f, Supplementary Fig. 4a, and Supplementary Data 3) co-culture with PBMCs. To further determine if the reprogramming to iMSCs or their expansion in culture may cause any chromosomal abnormalities, we performed digital karyotyping using SNP arrays. We did not identify any chromosomal abnormalities after either 1 week or 4 weeks of in vitro culture (Supplementary Figs. 5–7). These data demonstrated that human PBMCs can be efficiently reprogrammed into iMSCs using our nonintegrating episomal vector system.

**Reprogramming PBMCs into iMSCs without OCT4 or KLF4.** To assess the essentiality of the five factors, we performed reprogramming by omitting a single factor in separate experiments. PBMCs from various donors were used. Surprisingly, we found that skipping OCT4, a critical factor for blood cell reprogramming, still allowed the generation of a considerable number of MSC-like colonies (Fig. 3a and Supplementary Data 4). In

addition, PBMCs could be converted to iMSCs without KLF4, although at a ~35% decreased efficiency (Fig. 3a). Omitting SOX9 not only significantly reduced the number of colonies formed but the reprogrammed cells were round in shape instead of spindle-like MSCs suggesting that SOX9 played a pivotal role in determining the MSC fate (Supplementary Fig. 2b). By comparison, hardly any colonies were formed in the absence of BCL-XL or MYC. Taken together, SOX9, BCL-XL, and MYC were indispensable for reprogramming PBMCs into iMSCs.

The iMSCs generated with the three different combinations of reprogramming factors, 5F, 4FnoO (5F minus OCT4), and 4FnoK (5F minus KLF4), were morphologically similar: they were all spindle-shaped, resembling MSCs (Supplementary Fig. 2b). We evaluated the proliferation of the iMSCs generated from different conditions and compared it with primary human bone marrow MSCs (BMMSCs) (Supplementary Fig. 2c). Primary human BMMSCs showed slowed proliferation after ~1 month in culture. The iMSCs reprogrammed from PBMCs displayed an enhanced in vitro proliferative capacity compared with BMMSCs. While the 5F iMSCs and 4FnoK iMSCs have similar proliferation ability, the 4FnoO iMSCs showed slower proliferation compared with the other two types of iMSCs (5 F iMSCs and 4FnoK iMSCs). More than 100-fold more 5F iMSCs were generated than the human primary BMMSCs after ~1 month culture. In addition, >90% of the reprogrammed cells expressed the MSC marker CD73 (Fig. 3b) 4 weeks after vector transfection. To monitor the reprogramming process in more detail, we evaluated the expression of the MSC markers CD73 and CD90 at 2-, 3-, and 4-week post-transfection (Fig. 3c). We found that more than 60% of cells reprogrammed from either 5F or 4FnoK conditions became CD90+ by week 2, whereas only ~6% of cells from 4FnoO were CD90+, suggesting that OCT4 promoted the formation of CD90+ cells.

**Omission of OCT4 impaired the differentiation potential of iMSCs.** A characteristic feature of MSCs is the potential for trilineage differentiation into osteoblasts, adipocytes, and

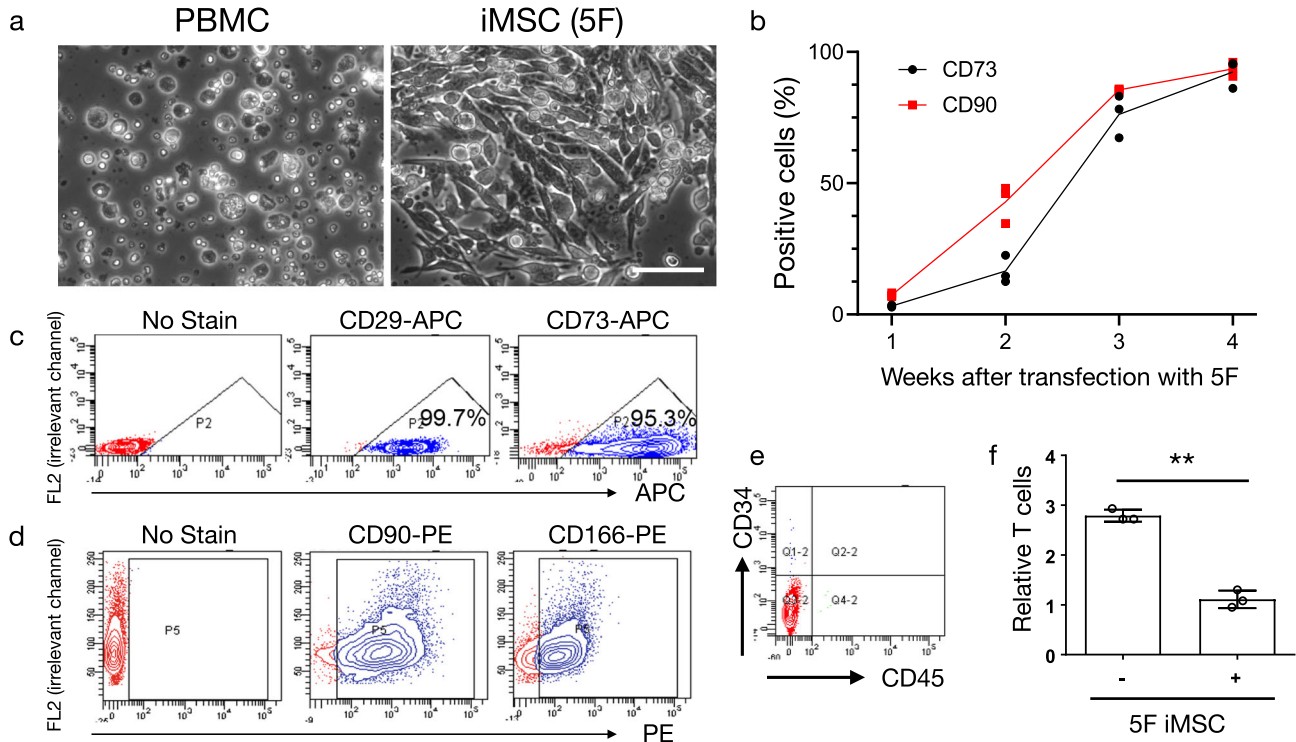

**Fig. 2 Direct reprogramming of human PBMCs into iMSCs using five episomal vectors expressing OCT4, BCL-XL, MYC, KLF4, and SOX9 (5F).**
**a** Representative images of human PBMCs and iMSCs 14 days after reprogramming with five factors (5F). Scale bar represents 100 μm. **b** Changes in the percentage of cells expressing the MSC markers CD73 and CD90 as measured by flow cytometry of 5F-transfected PBMCs over time. **c, d** Flow cytometry plots of typical MSC marker expression (CD29, CD73, CD90, CD166) at 4 weeks after reprogramming. $n = 3$ biologically independent samples for time point. **e** Blood cell markers (CD45 and CD34) were assessed 4 weeks after transfection of reprogramming factors. **f** iMSCs significantly inhibited T-cell proliferation after 3 days of co-culture with PBMCs. **P = 0.0007. Error bars indicate standard deviation. $n = 3$ biologically independent samples for each group.

chondrocytes[20]. To assess the functionality of iMSCs reprogrammed with 5F, 4FnoO, or 4FnoK, we cultured iMSCs in three lineage-specific induction media, followed by RT–qPCR analysis on the marker genes of osteogenesis, adipogenesis, and chondrogenesis.

The expression levels of runt-related transcription factor 2 (*RUNX2*), an early marker of osteogenic commitment, as well as the later osteogenic markers *SP7* and alkaline phosphatase (*ALP*), were significantly decreased in the 4FnoO-reprogrammed iMSCs compared with 5F- or 4FnoK-reprogrammed iMSCs ($P = 0.01$ and 0.03, respectively; Tukey's multiple comparisons test, Fig. 3d and Supplementary Data 5). To confirm the osteogenic commitment, we assessed calcium deposits by Alizarin Red S staining. Mineralization was observed in iMSCs reprogrammed with either 5F or 4FnoK but not in 4FnoO-reprogrammed iMSCs (Fig. 3g).

Regarding chondrogenic differentiation, there was no significant difference in the expression of chondrogenic marker genes such as *ACAN* among the three groups (Fig. 3e and Supplementary Data 5). Alcian blue staining, which stains for aggrecans associated with MSC chondrogenic potential, also showed no significant difference among the three groups (Fig. 3g). However, SOX9 expression was significantly reduced in 4FnoO iMSCs (4FnoO vs. 4FnoK, $P = 0.005$). These data suggested that omitting OCT4 also impaired the chondrogenic differentiation potential of iMSCs. Taken together, these five factors were necessary for the generation of iMSCs with unbiased differentiation potential. Conversely, reprogramming without OCT4 led to the formation of dysfunctional iMSCs.

After the induction of adipogenic differentiation, lipoprotein lipase (*LPL*) and fatty acid-binding protein 4 (*FADP4*) were expressed at substantially lower levels in 4FnoO iMSCs than in either 5F or 4FnoK iMSCs (Fig. 3f and Supplementary Data 5). We used Oil Red O staining to visualize lipid droplets in functional adipocytes. Consistent with the adipogenic gene expression data, iMSCs reprogrammed without OCT4 failed to differentiate into functional adipocytes (Fig. 3g). Of interest, omitting KLF4 led to the expression of higher levels of adipocyte markers and the formation of larger oil droplets, suggesting that KLF4 played a role in restricting adipogenic-biased MSCs.

To evaluate the immunomodulatory potentials of iMSCs reprogrammed with 5F, 4FnoO, or 4FnoK, we compared a list of major immunoregulatory cytokines, chemokines, and soluble factors secreted by MSCs[21,22] using the normalized gene counts from the RNA-seq data (Supplementary Fig. 4b and Supplementary Data 6). We found that compared with 5F iMSCs, in addition to impaired trilineage differentiation potential, the 4FnoO iMSCs showed significantly reduced gene expression on many immunoregulatory cytokines/chemokines, such as *IL-10*, *HGF*, *VCAM1*, *CCL2*, *CXCL14* (Supplementary Fig. 4b). Both 5F and 4FnoK iMSCs showed comparable levels of immunoregulatory cytokines/chemokines gene expression compared to the primary human bone marrow-derived MSCs[23].

**Global gene expression analysis of iMSCs reprogrammed from PBMCs.** To investigate the mechanisms underlying the distinct features of iMSCs reprogrammed with different factors (i.e., 5F, 4FnoO, and 4FnoK), we conducted transcriptome analysis 4 weeks after reprogramming factor transfection. We chose 4 weeks because >90% of the reprogrammed cells expressed MSC

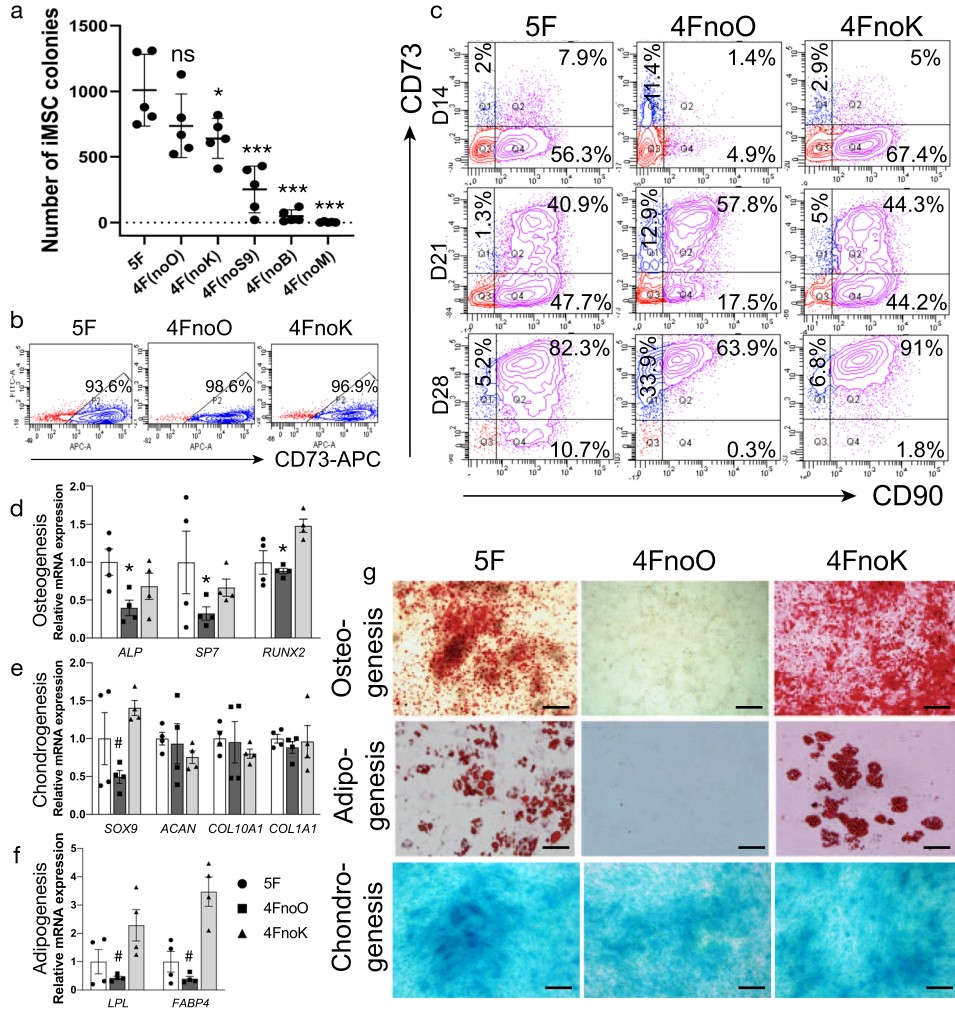

**Fig. 3 Impaired trilineage differentiation potential when reprogramming without OCT4. a** Reprogramming efficiency with the five-factor combination and removing one of the five factors. One-way ANOVA and Dunnett's multiple comparisons test, *$P < 0.05$ vs. 5F group, ***$P < 0.001$ vs. 5F group. ns: not significant. Error bars indicate standard deviation. $n = 5$ for each group from biological independent donors. **b** Flow cytometry analysis of the MSC marker CD73 4 weeks after transfection with 5F, 4FnoO (no OCT4), and 4FnoK (no KLF4). **c** Flow cytometry analysis of the MSC markers CD73 and CD90 at 2, 3, and 4 weeks after transfection with 5F, 4FnoO (no OCT4), or 4FnoK (no KLF4). **d–f** RT–qPCR analysis of osteogenesis-, adipogenesis-, and chondrogenesis-related genes in iMSCs reprogrammed with 5F, 4FnoO, and 4FnoK 2 weeks after multilineage differentiation. Tukey's multiple comparisons test, *$P < 0.05$, 4FnoO vs. 5 F and 4FnoK group. #$P < 0.05$, 4FnoO vs. 4FnoK group. $n = 4$ biologically independent samples for each group. Error bars indicate standard deviation (SD). **g** Multilineage differentiation of iMSCs reprogrammed with 5F, 4FnoO, or 4FnoK. Cells were cultured in osteogenic, adipogenic, or chondrogenic induction medium for 2–4 weeks and stained with Alizarin Red (osteogenesis), Oil Red O (adipogenesis), or Alcian blue (chondrogenesis), respectively. Scale bars represent 200 μm.

markers at this time point, and the nonintegrating episomal viral vectors were cleared from the reprogrammed cells[7]. First, we investigated the differentially expressed genes (DEGs) between the 5F, 4FnoO, or 4FnoK iMSCs. DEG analysis identified 827 significantly down- and 538 significantly upregulated genes in 4FnoO iMSCs compared to 5F iMSCs (FDR < 0.05 and fold change (FC) > 2, Fig. 4a and Supplementary Data 7). Of note, 5F and 4FnoK iMSCs showed similar transcriptomes with only 24 DEGs, consistent with their seemingly identical differentiation potentials (Supplementary Fig. 8). Hierarchical clustering analysis identified a set of genes highly enriched in 5F and 4FnoK iMSCs, some of which were reported as MSC lineage signature genes, such as *SRPX*, *S1PR3*, *ROBO2*, *NCAM1*, *COL5A1*, and *COL4A1* etc[24–26] (Fig. 4b and Supplementary Data 7). Furthermore, the 4FnoO iMSCs displayed a significant decrease in the expression of mesoderm-regulating genes, including *SOX4*, *SALL4*, and *TWIST1* (Supplementary Data 7). We speculated that these downregulated genes might be associated with the

impaired functionality of 4FnoO iMSCs. We then performed Gene Ontology (GO) enrichment analyses to explore the pathways associated with genes expressed at low levels in 4FnoO iMSCs. We found that 1365 DEGs were enriched in the biological processes of axonogenesis, extracellular structure organization, ossification, and cartilage development (Fig. 4c). The top identified Kyoto Encyclopedia of Genes and Genomes (KEGG) pathways were the PI3K-Akt signaling and calcium signaling pathways (Fig. 4d). These data helped explain the functional defects in osteogenesis of 4FnoO iMSCs and further understanding of the role of OCT4 in reprogramming PBMCs into iMSCs.

To compare the iMSCs reprogrammed from PBMCs with primary human MSCs, we downloaded RNA-seq data generated from primary human bone marrow-derived MSCs (BMMSC)[23] and primary human adipose-derived MSCs (AdMSC). First, we analyzed the transcriptional similarity of the iMSCs in our study to the primary human MSCs using principal component analysis

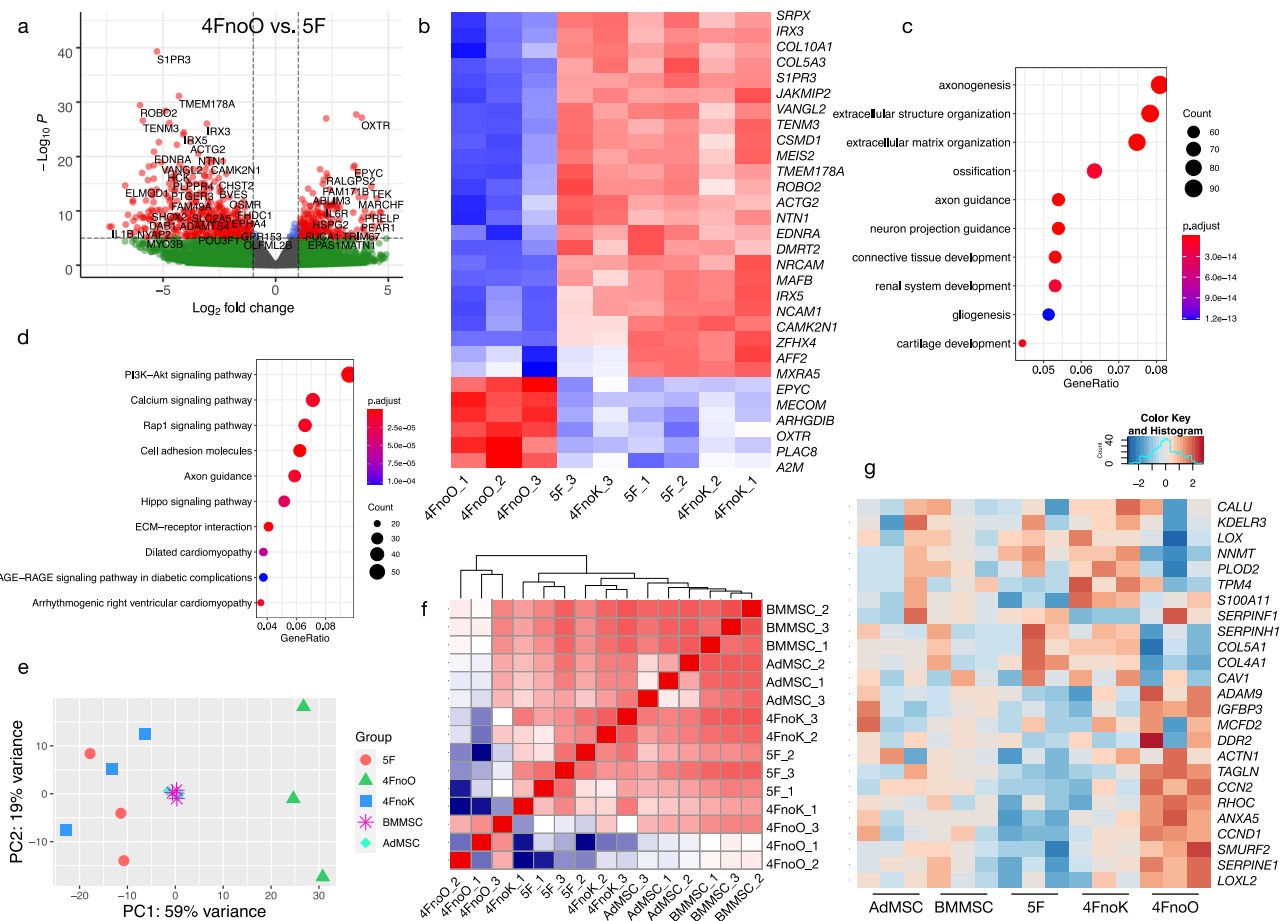

**Fig. 4 Comparative transcriptome analysis of iMSCs reprogrammed from PBMCs. a** Volcano plot showing differentially expressed genes identified in 4FnoO iMSCs compared with 5F iMSCs. Each dot represents a gene. The red dots are genes significantly upregulated (right) or downregulated (left) in 4FnoO iMSCs (Cutoff: $P < 10e^{-6}$, fold change > 2). **b** Heatmap showing the top 30 differentially expressed genes between 5F iMSCs and 4FnoO iMSCs (ranked by $p$-value). **c**, **d** Dot plots showing the top Gene Ontology (GO) biological process (BP) terms (**c**) and KEGG pathways (**d**) enriched from DEGs in 4FnoO iMSCs compared to 5F iMSCs. **e** PCA of RNA-seq from iMSCs 4 weeks after reprogramming with 5F, 4FnoO, or 4FnoK, primary human bone marrow-derived MSCs (BMMSC) and primary adipose-derived MSCs (AdMSC). For each condition, iMSCs were reprogrammed from PBMCs derived from three biologically independent donors. **f** Pearson correlation analysis of iMSCs and primary MSCs. **g** Comparison of twenty-four genes previously determined to be specific to the MSC lineage between primary MSCs and iMSCs.

(PCA) (Fig. 4e). The reduction of the multi-dimensional dataset into two principal component (PC) dimensions enables the unbiased comparison and visualization of the transcriptomes between samples. As expected, the results showed that 4FnoO iMSCs were distinct from the other two iMSC groups (Fig. 4e), consistent with the impaired differentiation potential of 4FnoO iMSCs when compared with 5F and 4FnoK iMSCs. The transcriptomes of human BMMSC and AdMSC were very similar to each other. Furthermore, the variation captured in PC1 demonstrated closer similarity of 5F and 4FnoK iMSCs with the primary MSCs compared to 4FnoO iMSCs, which tended to cluster further away from BMMSC and AdMSC (Fig. 4e). Pearson correlation analysis confirmed that the 4FnoK and 5F iMSCs retained strong transcriptome correlation with the primary MSCs, while the 4FnoO iMSCs had less correlation with the primary MSCs (Fig. 4f). A panel of 24 MSC lineage genes[25,26] were compared between the primary MSCs and our iMSCs (Fig. 4g). The 4FnoO iMSCs showed distinct expression patterns of these MSC signature genes that contrasted strongly with other groups. Noteworthy is that *COL4A1*, *COL5A1*, *LOX*, *NNMT*, which are

known to be upregulated in MSCs versus fibroblasts[24,27], were downregulated in 4FnoO iMSCs.

**OCT4 increased the chromatin accessibility during PBMC reprogramming into iMSCs.** Genome-wide chromatin accessibility can provide mechanistic insights at the molecular level into cell fate decisions, especially during the reprogramming process. Thus, we performed ATAC-seq[28] analysis on iMSCs 4 weeks after reprogramming PBMCs with 5F, 4FnoO, or 4FnoK. Open chromatin regions were identified as peaks in the ATAC-seq dataset. Furthermore, after peak calling, the relative genomic distribution of ATAC peaks showed reduced peaks within promoter regions in iMSCs generated without OCT4 (Fig. 5a). In contrast, these cells had more open chromatin at intron regions. These results suggested that OCT4 may preferentially bind promoter regions to promote chromatin accessibility during reprogramming.

Similar to what was observed in the RNA-seq transcriptomic data, PCA of normalized ATAC-seq read counts showed that chromatin accessibility of three groups of iMSCs (5F, 4FnoO, and

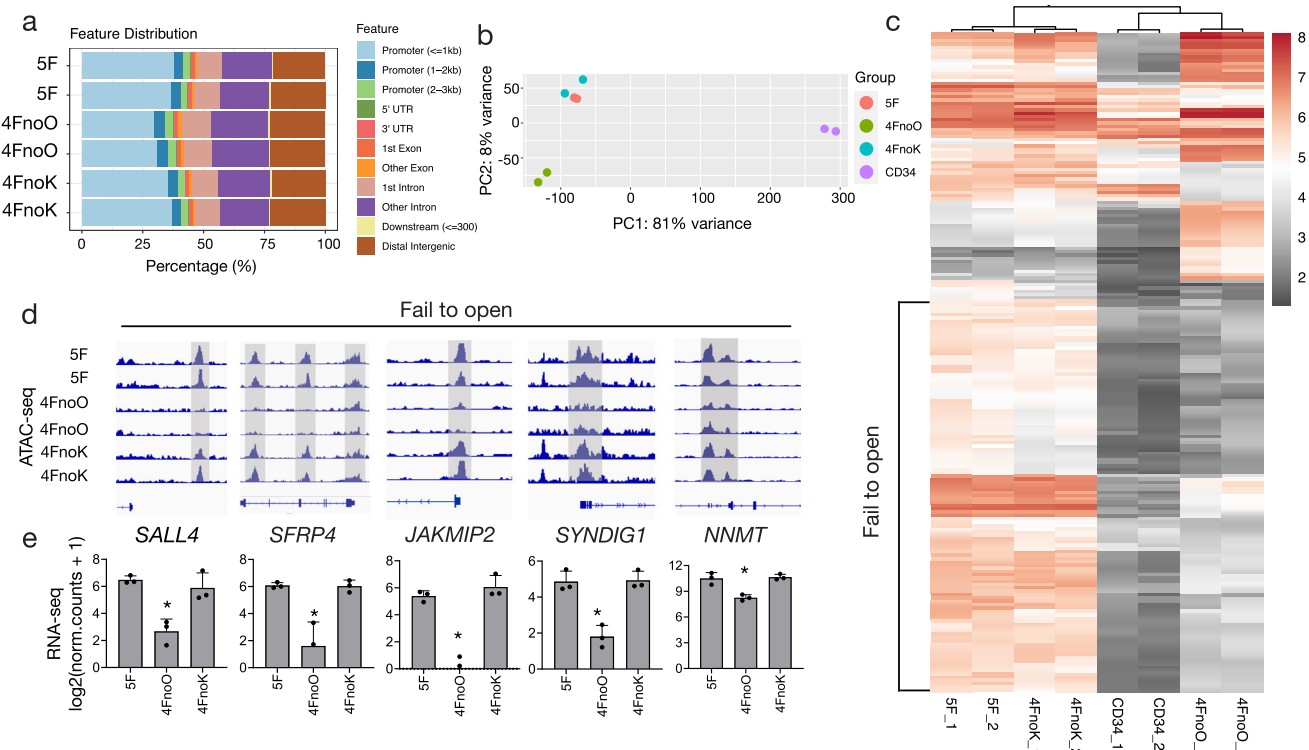

**Fig. 5 Chromatin accessibility analysis of iMSCs reprogrammed with different factor combinations. a** Genomic location of ATAC-seq peaks from 5F, 4FnoO, and 4FnoK iMSCs. **b** PCA using normalized ATAC-seq counts from 5F, 4FnoO, and 4FnoK iMSCs, and two datasets from bone marrow-derived CD34+ cells (SRR2920489 and SRR2920490). For each condition, the chromatin accessibility was profiled from iMSCs that were reprogrammed from two biologically independent donors. **c** Heatmap showing ATAC-seq signals with the top 200 most different peaks (ranked by padj). Red represents chromatin regions with more mapped reads, suggesting possible chromatin openness. Gray represents chromatin regions with fewer mapped reads, suggesting closed chromatin. **d** Selected genomic views of the ATAC-seq data using IGV (2.8) for the indicated groups. For each gene, all genome views are on the same vertical scale. **e** The bar plot showing RNA-seq gene expression values for the respective genes shown above in the genome view. RNA-seq gene expression levels are shown as $\log_2()$ normalized read counts. $n = 3$ biologically independent samples for each group. *$P \leq 0.05$; error bars indicate standard deviation.

4FnoK) were well-separated from each other, in which the accessible chromatin regions were mainly different in 4FnoO cells (PC1 = 52% variance, Supplementary Fig. 9). However, in contrast to the similar transcriptomes between 5F and 4FnoK iMSCs (Supplementary Fig. 8 and Fig. 4a), ATAC-seq analysis showed that there was a clear separation between 5F and 4FnoK iMSCs (PC2 = 19%, Supplementary Fig. 9). These data suggested that both OCT4 and KLF4 facilitate chromatin remodeling during reprogramming. To compare the changes in chromatin accessibility during reprogramming, we downloaded the ATAC-seq data of primary CD34+ cells from bone marrow (SRR2920489, SRR2920490)[29], which are similar to our reprogramming-initiating cells in this study. The datasets were processed using the same analysis pipeline. PCA revealed that CD34+ hematopoietic progenitor cells clustered separately from the three groups of reprogrammed iMSCs (Fig. 5b), whereas 5F iMSCs and 4FnoK iMSCs were clustered closely with each other.

We also noticed that some chromatin regions remained closed in both CD34+ and 4FnoO iMSCs, whereas the same regions were in an open configuration in the 5F and 4FnoK iMSCs (Fig. 5c). These data suggested that OCT4, but not KLF4, played a critical role in opening chromatin during the reprogramming process. More specifically, OCT4 opened the chromatin of the stemness-associated gene *SALL4*, Wnt signaling-related genes such as *SFRP4*, microtubule-binding and glutamate receptor binding-related genes *JAKMIP2* and *SYNDIG1*, and MSC lineage signature gene *NNMT* (Fig. 5d). These genes with reduced

ATAC-seq peaks in 4FnoO iMSCs also showed significantly reduced mRNA expression, indicating a consistency between transcriptome and chromatin accessibility data (Fig. 5e and Supplementary Data 8).

**OCT4 induced a global hypomethylation during reprogramming.** DNA methylation is the most common epigenetic modification of the genome to control gene expression in mammalian cells[30] and the differentiation or self-renewal of MSCs[13]. To determine the effects of reprogramming factors on methylation levels and patterns in iMSCs, we assessed genome-wide CpG methylation profiles in 5F, 4FnoO, and 4FnoK iMSCs at week four using RRBS. First, we profiled CpG methylation patterns on five different genomic features (all sites, promoters, exons, introns, and transcription start sites (TSSs) (Fig. 6a, b and Supplementary Data 9). We found that iMSCs reprogrammed without OCT4 showed a globally hypermethylated CpGs compared to iMSCs reprogrammed with OCT4 (Fig. 6a, b). Specifically, when reprogramming in the absence of OCT4, we identified 10,760 differentially methylated cytosines (DMCs) (20%, $q = 0.1$, Supplementary Data 10), of which 9004 DMCs were hypermethylated and 1756 DMCs were hypomethylated (4FnoO vs. 5F). Among these sites, 7.7% were within promoter regions, and 7.9% were within exon regions (Fig. 6c). In contrast, there was no significant difference in CpG methylation within all five genomics features in the iMSCs when reprogrammed in the absence of

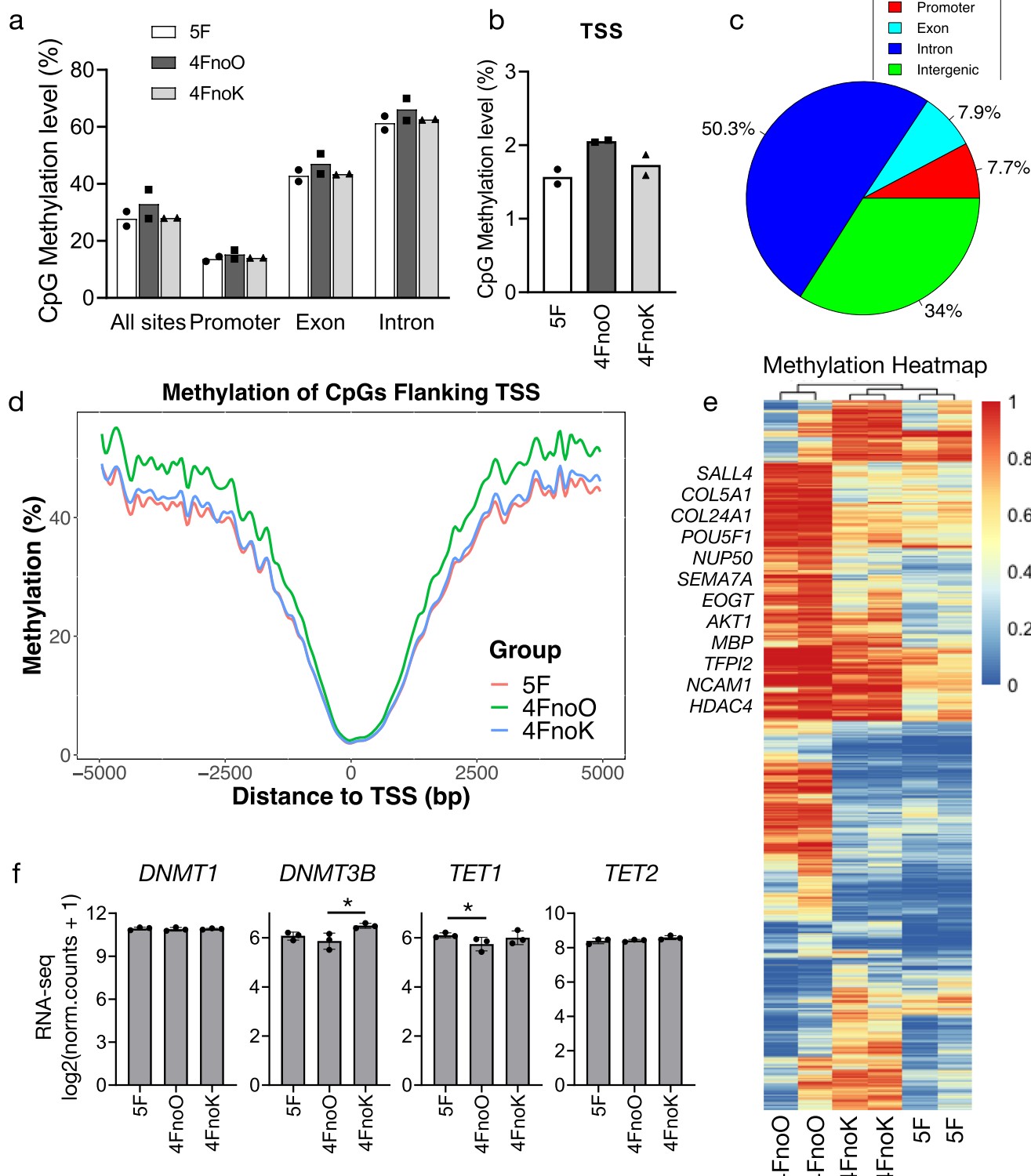

**Fig. 6 Genome-wide DNA demethylation induced by OCT4 during PBMC reprogramming. a** The bar graph showing the methylation levels of all sites, promoters, exons, and intron regions from 5F, 4FnoO, and 4FnoK iMSCs. $n = 2$ biologically independent samples for each group. **b** The methylation levels of the TSS region. $n = 2$ biologically independent samples for each group. **c** The percentage of differentially methylated CpGs (DMCs) between 5F and 4FnoO iMSCs annotated within the promoter, exon, intron, and intergenic regions shown in the pie chart. **d** The average methylation levels surrounding the TSSs (−5000 to +5000 bp) in 5F, 4FnoO, and 4FnoK iMSCs. **e** Hierarchical clustering and heatmap analysis of 13,974 DMCs. **f** The bar plot showing the log$_2$() normalized read counts from RNA-seq. $n = 3$ biologically independent samples for each group. *$P < 0.05$; error bars indicate standard deviation.

KLF4 (Fig. 6a, b). Of the 3849 CpG sites significantly different (20%, $q = 0.1$) between the 5F and 4FnoK groups, 3698 CpG sites were hypermethylated, and 151 sites were hypomethylated. When measuring the average methylation against the distance to the TSS, there was a global hypermethylation pattern in the iMSCs reprogrammed without OCT4 (Fig. 6d, $p < 0.0001$), suggesting that OCT4 was critical for global demethylation during reprogramming of PBMCs to iMSCs.

We performed hierarchical clustering on six RRBS datasets and generated a heatmap using the beta value of all common CpG sites. As expected, two datasets from 4FnoO clustered together, enriched a set of hypermethylated DMCs that were not observed in the 5F and 4FnoK datasets (Fig. 6e). Since the cells reprogrammed from 5F and 4FnoK were very similar in their transcriptomes, chromatin openness, and methylation levels, we focused on our comparisons in the iMSCs programmed using 5F vs. 4FnoO. We annotated 10,760 DMCs and identified 665 differentially methylated genes (DMGs) between 5F and 4FnoO iMSCs (Supplementary Data 10) which were subject to GO enrichment analysis (Supplementary Fig. 10). Similar to the GO enrichment analysis based on RNA-seq data, DMGs were enriched in axonal guidance signaling and mesenchyme development. Of note, *POU5F1*, *SALL4*, *NCAM1*, *HDAC4*, and MSC lineage signature gene *COL5A1* were significantly hypermethylated in iMSCs reprogrammed using 4FnoO compared with the iMSCs programmed using 5F (Supplementary Data 10), suggesting that these genes might be associated with the impaired functionality in the 4FnoO iMSCs.

Demethylation may occur passively. DNMT1 is the most abundant DNA methyltransferase in mammalian cells and is considered the key methyltransferase responsible for DNA methylation maintenance, and its inhibition will result in passive demethylation. We found that the expression levels of *DNMT1* in iMSCs reprogrammed with or without OCT4 were similar (Fig. 6f and Supplementary Data 8), suggesting minimal role of *DNMT1* in OCT4-mediated demethylation. We then suspected that active DNA demethylation might have contributed to the global hypomethylation. Active DNA demethylation is mainly regulated by ten-eleven translocation (TET) enzymes[31]. We observed that the expression of *TET1*, but not *TET2*, was significantly reduced when reprogramming without OCT4 (Fig. 6f), suggesting that *TET1* might have contributed to OCT4-induced global demethylation. Meanwhile, the expression level of *DNMT3B* was significantly increased when reprogramming without KLF4, suggesting a role of KLF4 in regulating DNA methylation homeostasis via de novo DNA methyltransferase DNMT3B (Fig. 6f).

**Integrated analysis of multiomics data**. To assess the influence of methylation on gene expression, we performed integration analysis of DMGs and DEGs datasets. We found the co-occurrence of 67 genes between 5F and 4FnoO iMSCs (Fig. 7a and Supplementary Table 1). Hypergeometric test was applied to show that the overlap is significant. Our analysis suggested that the observed difference in functionality between 5F and 4FnoO iMSCs might be a consequence of the difference in the methylation status of these 67 genes. Among these genes, *ZFHX4*, *SLC8A2*, *NCAM1*, *TFPI2*, and *SALL4* were the most differentially expressed (Fig. 7b). When PBMCs were reprogrammed without OCT4, not only were these genes significantly hypermethylated on either promoters or exons compared to PBMCs reprogrammed with OCT4 (Supplementary Data 10), but some chromatin regions of these genes also remained inaccessible/closed (Fig. 7c). Consistent with the hypermethylation of the four genes, their transcription levels were close to zero (Fig. 7d and Supplementary Data 8).

ZFHX4, a transcription-related zinc finger protein involved in the mesodermal commitment pathway, is upregulated in both embryonic stem cell-derived and bone marrow (BM)-derived MSCs[32,33]. These reports, together with our findings, indicate that ZFHX4 may serve as an MSC marker. In addition, neural cell adhesion molecule (NCAM), also called CD56, is expressed on human MSCs and was proposed as a marker for human MSC isolation[34,35]. Also, CD56+ cells showed increased colony formation ability, suggesting CD56 expression enriches MSCs with self-renewal potency[36]. On the other hand, BM-MSCs from NCAM-deficient mice exhibited defective migratory ability and significantly impaired adipogenic and osteogenic differentiation potential[37].

Many genes have been proposed as MSC surface marker genes, but no consensus has been reached yet. To screen possible trilineage differentiation function associated MSC markers, we compared ten well-established MSC surface markers between primary MSCs and our iMSCs (Fig. 7e and Supplementary Fig. 11). We found that other than NCAM1, four additional MSC surface markers (CD90, PDGFRB, CD82, and FZD5) were highly expressed in both primary MSCs and 5F/4FnoK iMSCs but downregulated in 4FnoO iMSCs (Fig. 7e). Taken together, integrated analysis of multiomics data lead to the identification of putative functional MSC markers, and our dataset enables the mining for additional MSC surface markers that co-associate with functional potential.

## Discussion

In order to generate integration-free human iMSCs, we reprogramed human PBMCs into iMSCs directly using nonintegrating episomal vectors in this study. This approach successfully generated iMSCs from peripheral blood CD34+ cells using five factors (OCT4, MYC, SOX9, KLF4, and BCL-XL). The reprogrammed cells expressed typical MSC markers and had trilineage differentiation potential. Conversely, omitting OCT4 led to the formation of iMSCs with impaired differentiation capacity. Mechanistically, we found that OCT4 played a critical role in opening chromatin and demethylating lineage-specific gene loci.

OCT4 is a pivotal reprogramming factor for converting somatic cells to induced pluripotent stem cells[38]. Previously, we reported that this factor could reprogram human cord blood CD34+ cells directly into iMSCs in a dose-dependent manner[11]. This study found that OCT4 was indispensable for generating iMSCs with multilineage differentiation ability. OCT4 is a pioneer factor that binds and possibly opens up closed chromatin during human pluripotency reprogramming[39]. Our results suggested that SOX9, MYC, and KLF4 were not responsible for the observed chromatin changes during the initial reprogramming stage, whereas OCT4 was crucial for global demethylation and activation of stemness genes. Furthermore, our ATAC-seq data showed that only OCT4 could open certain chromatin regions inaccessible in HSPS CD34+ cells. OCT4 was also reported to be a key transcription factor for the self-renewal and survival of MSCs[40,41]. Together, OCT4 is pivotal in reprogramming by opening closed chromatin, facilitating cell fate conversion.

Reprogramming to pluripotency followed by differentiation to MSCs may lead to trace amount of undifferentiated iPSCs in the end products, whereas direct reprogramming of hematopoietic cells to iMSCs virtually abolished this possibility. Apart from BCL-XL, which improves blood cell survival[16,42], our reprogramming factor combination was similar to the original Yamanaka factors. The key difference was the replacement of SOX2 with SOX9. Although the SOX family of genes has similar DNA binding characteristics, replacing SOX2 with other SOX factors, such as SOX7 or SOX17, demolished the activity of the Yamanaka

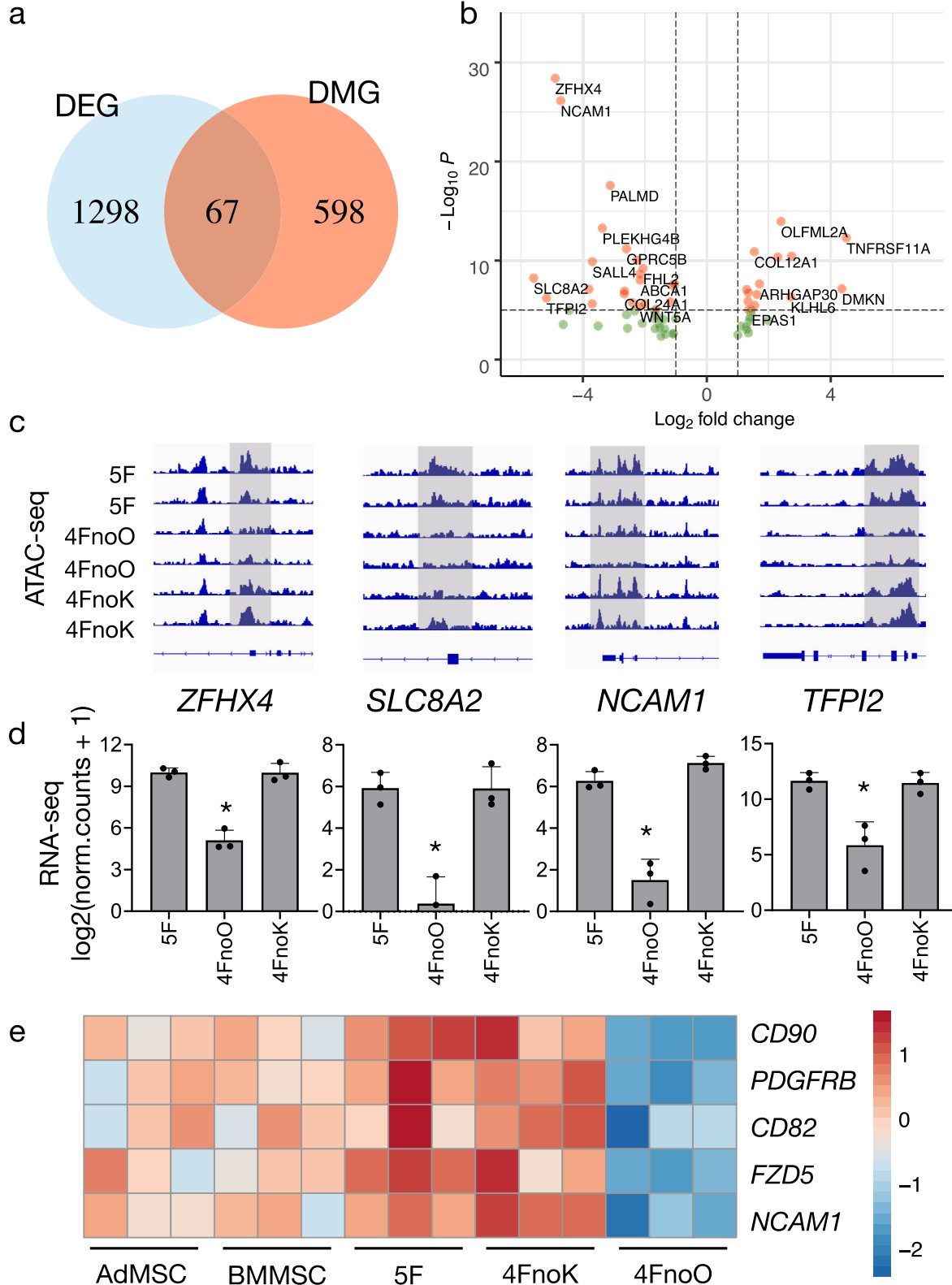

**Fig. 7 Integrated analysis of multiomics data identifies putative MSC markers. a** Venn diagram illustrating the overlap between the differentially expressed genes (DEGs) and differentially methylated genes (DMGs) between 5F iMSCs and 4FnoO iMSCs. A total of 1365 DEGs and 665 DMGs were identified; 67 of these were both differentially expressed and differentially methylated. **b** Volcano plot showing 67 overlapping genes between the DEG and DMG. pCutoff = 10e−6, log₂ FC > 1). **c** Selected genomic views of the ATAC-seq data using IGV (2.8) for the indicated groups. For each gene, all genome views are on the same vertical scale. **d** The bar plot showing the RNA-seq gene expression values for the respective genes, which are shown above in the genome view. RNA-seq gene expression levels are shown as log₂() normalized read counts. *, $P < 0.05$; error bars indicate standard deviation. $n = 3$ biologically independent samples for each group. **e** Heatmap showing the normalized gene read count after log₂() transformation from RNA-seq.

reprogramming cocktail[43]. Here, we chose SOX9 because it is an essential transcription factor that controls the development of the musculoskeletal system[44]. Inactivation of Sox9 in limb buds before mesenchymal condensations led to a complete absence of both cartilage and bone[45]. As expected, replacing SOX2 with SOX9 not only prevented the generation of TRA-1-60 positive iPSCs but significantly increased the number of iMSC-like colonies. These data suggest that unlike SOX2, which enables reprogramming of PBMCs to the pluripotent stage, SOX9 likely restricts cells to the mesodermal lineage.

DNA methylation is a key barrier to cellular reprogramming[46,47]. The reprogramming process involves the erasure and re-establishment of methylation patterns contributing to the silencing of signature genes characteristic of the parental cell identity and activation of the signature genes essential to the identity of the reprogrammed daughter cells. This is analogous to the global DNA demethylation in primordial germ cells and zygotic paternal pronuclei. After fertilization, global DNA demethylation occurs rapidly in the zygote, with the paternal genome being actively demethylated while the maternal genome loses its methylation marks passively[48]. Here, we reported that OCT4 was critical for global demethylation and the establishment of an MSC-specific epigenetic landscape. A recent study suggested that proper *TET1* expression during reprogramming is essential for epigenetic rewiring to generate high-quality iPSCs[49]. Dysfunction in TET1-mediated DNA demethylation during reprogramming causes insufficient reprogramming with epigenetic abnormalities. Since *TET1* is a target gene of OCT4, positive feedback between OCT4 and TET1 may partially explain the methylation differences between the 5F and 4FnoO iMSCs.

Great efforts have been made to identify specific markers for MSCs. The clinical application of MSCs is hampered by the absence of one or more molecular signatures indicative of their functionality. In 2006, the International Society for Cellular Therapy proposed minimal criteria for MSCs, in which CD105, CD73, and CD90 were considered MSC surface markers[50]. However, these markers do not predict the multipotent functionality of MSCs[51]. In our study, we found that although the iMSCs programmed using 4FnoO expressed CD73 and CD90, these cells showed significantly impaired multilineage differentiation potential. Nevertheless, our multiomics analysis on reprogrammed iMSCs with distinct functionality may facilitate the identification of functional MSC markers. When compared 10 well-established MSC surface markers between primary MSCs and our iMSCs, we found that six of them were either comparable or even increased in 4FnoO iMSCs, including CD271, CD105, CD73, CD166, CD146, and PODXL (Supplementary Fig. 11). This data strongly suggests that these MSC marker genes might not truly correlate with the functionality of iMSCs. On the other hand, we also identified a group of genes from the DEG list, including CD90, PDGFRB, LIFR (CD118), CD82, FZD5, and NCAM1 (CD56), whose expression positively correlated with multilineage differentiation potential. CD90 and PDGFRB are well-known MSC markers[34,50], both of which have been directly associated with MSC therapeutic functions. Furthermore, the tetraspanin family member CD82 has been proposed as a MSC marker[52]. As an essential receptor in mediating Wnt/b-catenin signaling, FZD5 is highly expressed in human regenerative MSCs[53,54]. Here, we found that iMSCs reprogrammed using 4FnoO, which had impaired multipotency, showed a 500-fold reduction in NCAM1 mRNA expression, inaccessible NCAM1 chromatin regions remained, and a significantly hypermethylated promoter region. This is consistent with previous reports that NCAM1 was highly expressed in primary MSCs[55] and was associated with a cell population with greater clonogenic potential[35]. Taken together, it is tempting to speculate that CD56, CD82, and CD118 may be considered as MSC markers in future investigations.

In summary, we have identified a combination of five factors for efficient reprogramming of PBMCs into iMSCs. We also identified OCT4 as a critical gene for generating functional iMSCs by opening closed chromatin and inducing global demethylation. Multiomics analysis of reprogrammed iMSCs led to the discovery of multiple putative functional MSC markers that may be used to predict therapeutic activity and aid in advancing the application of MSCs to clinical regenerative medicine.

## Methods

**Episomal vectors.** All the episomal vectors used in this study were prepared as previously described[9,16,56]. Briefly, plasmids OCT4 (O), BCL-XL (B), MYC (M), KLF4 (K), SOX2 (S2), and SOX9 (S9) were constructed by inserting the open reading frames of these genes into our improved ENBA/oriP-based episomal vector, in which the SFFV promoter and Wpre element were included to drive the high-level transgene expression in hematopoietic cells[11].

**Human PBMCs isolation.** The study was approved by the Institutional Review Board of the Loma Linda University. Human peripheral blood was obtained from a local blood donation center (LifeStream, San Bernardino, CA) without receiving individually identifiable information. PBMCs were isolated from peripheral blood by standard density gradient centrifugation with Ficoll-Hypaque (1.077 g/mL) (G&E Healthcare; 17-1440-03) as previously described[16].

**Reprogramming of PBMCs to iMSCs.** PBMCs were cultured in erythroid medium composed of Stemline II Hematopoietic Stem Cell Expansion Medium (Sigma; S0192) supplemented with 100 ng/ml stem cell factor (Peprotech; 300-07), 10 ng/ml interleukin-3 (Peprotech; AF-200-03), 2 U/ml erythropoietin (Peprotech; 100-64), 20 ng/ml insulin growth factor-1 (Peprotech; 100-11), 1 mM dexamethasone (Sigma; D4902), and 0.2 mM 1-thioglycerol (Sigma; M6145). After 6 days of culture, $2 \times 10^6$ cells were transfected with the indicated plasmids by electroporation using the Amaxa Human CD34$^+$ cell nucleofector kit (Lonza; VPA-1003) and the program U-008 on an Amaxa Nucleofector II, according to the manufacturer's instructions. Briefly, a 70 μl electroporation solution was prepared for each nucleofection, including 57.4 μl of nucleofector solution, 12.6 μl of supplement, and 1 μg of each plasmid, as indicated for each group. After nucleofection, the cells were seeded onto fibronectin-coated plates in a mixture of erythroid medium and MSC medium (50:50 ratio) supplemented with small molecules, including 3 μM CHIR99021, 10 μM forskolin, 10 μM ALK inhibitor (SB431542), and 5 μM tranylcypromine hydrochloride, for two days and replaced with MSC medium after that. MSC-like colonies were formed 1 week after nucleofection. Two weeks after nucleofection, cells were cultured in MSC medium without adding small molecules. To calculate reprogramming efficiency, at day 14 after culture, the total number of iMSC colonies was divided by $1 \times 10^6$ PBMCs. For all subsequent experiments, including flow cytometry and sequencing, picking up individual colony was not performed. The cultures consist of pools from multiple colonies.

**Flow cytometry.** Cells were dissociated with Accutase (Innovative Cell Technologies, Inc., CA) and analyzed on a BD FACSAria II flow cytometer or Nanocellect WOLF cell sorter. To analyze iMSCs, cells were stained with hematopoietic markers CD34 and CD45, MSC markers CD73, CD90, CD29, and CD166, and the iPSC marker TRA-1-60, NANOG, and OCT3/4 with 1:100 dilution following the gating scheme provided in Supplementary Fig 12. All antibodies were purchased from ThermoFisher (San Diego, CA).

**In vitro differentiation of MSCs.** To evaluate iMSC function, we examined iMSC trilineage differentiation in vitro. For osteogenic differentiation, $2 \times 10^5$ cells were seeded in each well of a 6-well plate in α-MEM supplemented with 10% FBS, 0.1 μM dexamethasone, 200 μM ascorbic acid, 10 mM β-glycerol phosphate, 10 ng/ml BMP2 and 10 ng/ml BMP4. For adipogenic differentiation, $1 \times 10^5$ cells were seeded in each well of a 6-well plate in α-MEM supplemented with 10% FBS, 1 μM dexamethasone, 1 μM troglitazone, 10 μg/ml insulin, 0.5 mM iso-butylmethylxanthine, and 5 ng/ml FGF2. For chondrogenic differentiation, $4 \times 10^5$ cells were seeded in each well of a 6-well plate in α-MEM supplemented with 10% FBS, 0.1 μM dexamethasone, 200 μM ascorbic acid, 5.33 μg/ml linoleic acid, 0.35 mM L-proline, 10 ng/ml TGFβ3, 10 ng/ml TGFβ1, and 1% ITS.

All the culture media were changed every 2–3 days. Three to four weeks after the induction of differentiation, cells were fixed in 10% neutral buffered formalin for 5 minutes, followed by staining as previously described[11]. Briefly, Alizarin Red staining was performed to evaluate calcium deposits generated from osteoblasts, Oil Red O staining was performed to evaluate lipid droplets of adipocytes, and Alcian Blue staining was performed to evaluate mucopolysaccharides secreted by chondrocytes.

**Real-time qRT-PCR.** Total RNA was extracted using an RNA Isolation kit (BIOLINE, TN) with DNase I (Qiagen) treatment on the column. The RNA was

quantified using a NanoDrop 2000 (Thermo Scientific) and further diluted to equal concentrations. First-strand cDNA was synthesized using the High-Capacity cDNA Reverse Transcription kit (Thermo Fisher) according to the manufacturer's instructions. The sequences of primers for qPCR are listed in Supplementary Table 2. The qPCR was performed using SYBR Green PCR Master Mix (Applied Biosystems) and conducted on QuantStudio 7 Flex (Applied Biosystems by Life Technologies). Gene expression Ct value was normalized with ACTB, and relative quantitation of fold change was calculated using the 5F iMSCs group as reference.

**Digital karyotyping**. Genomic DNA was extracted from primary PBMCs and iMSCs cultured for 1 week (passage 3, P3) and 4 weeks (passage 10, P10), and DNA was hybridized to Infinium BeadChip (Illumina), followed by staining and scanning on an Illumina HiScan system. These arrays can interrogate 597,784 human SNP markers, thus yielding up to 50-fold better resolution (~100 kb) than conventional karyotyping by Giemsa banding. B allele frequency (BAF) and Log R ratio (LRR) were used to detect copy number variants (CNVs). When viewing the result, the blue points represent BAF, which is the proportion of hybridized sample that carries the B allele. The BAF values of 0.0, 0.5, and 1.0 for each locus (representing AA, AB, and BB) can be seen in a normal sample. The red points represent Log R ratio. Any deviations from 0.0 indicate copy number changes.

**MSC-T cell coculture assay**. To evaluate the immunomodulatory ability of iMSCs, human PBMCs were seeded at $4 \times 10^5$ cells in 1 ml MSC medium with Dynabeads Human T-Activator CD3/CD28 (Gibco). For co-culture of PBMCs and MSC cells, the MSC cells were seeded one day before culturing PBMCs. After 3 days of culture, T cells were stained with CD4-APC (BD) and CD8-PE (BD) at room temperature for 30 min and assessed by BD Canto II flow cytometry. For 6-day culture experiments, cells were passaged at 1:8 at day 3 and seeded in fresh MSC medium with or without MSC cells as indicated. On day 6, the CD4 positive and CD8 positive T cells were measured.

**Sample preparation for genomic analysis**. PBMCs from two to three individual subjects were used as biological replicates. Reprogramming of PBMCs was carried out using three different factor combinations: (1) 5F (OCT4, MYC, BCL-XL, SOX9, KLF4); (2) 4FnoO (MYC, BCL-XL, SOX9, KLF4); and (3) 4FnoK (OCT4, MYC, BCL-XL, SOX9). iMSCs were collected at the week four after PBMC reprogramming. iMSCs from each condition were divided into two aliquots: one aliquot was used to extract both RNA and DNA using the AllPrep DNA/RNA/miRNA Universal kit (Qiagen); the other aliquot was used for ATAC-seq analysis.

**RNA library construction and sequencing**. RNA was isolated using the AllPrep DNA/RNA/miRNA Universal kit (Qiagen), and libraries were constructed using the NuGEN Ovation Universal RNA-seq kit (TECAN). Briefly, 100 ng of total RNA was reverse transcribed and then converted into double-stranded cDNA (ds-cDNA) by adding DNA polymerase. The ds-cDNA was fragmented to ~200 bp using Covaris S220 and then underwent end repair to blunt the ends, followed by barcoded adapter ligation. The remainder of the library preparation followed the manufacturer's protocol. All libraries were quantified with a TapeStation 2200 (Agilent Technologies) and Qubit 3.0 (Life Technologies). The libraries were sequenced on an Illumina HiSeq 4000 (Illumina, San Diego) for 100 bp single-end sequencing. Detailed sequencing information is listed in the Supplementary Table 3.

**ATAC library construction and sequencing**. ATAC-seq libraries were constructed by following the improved protocol as previously described[57]. A total of 50,000 cells were used for each library construction. All libraries underwent five preamplification cycles after the transposition reaction, followed by three additional PCR amplification cycles that were determined by qPCR. Finally, the amplified libraries were purified using a Qiagen MinElute PCR Purification kit and quantified using the KAPA Library Quantification kit. The libraries were sequenced on an Illumina NextSeq 550 (Illumina, San Diego) with 75×2 paired-end sequencing. Detailed sequencing information is listed in the Supplementary Table 3.

**Reduced representation bisulfite sequencing library construction and sequencing**. gDNA was extracted using the AllPrep DNA/RNA/miRNA Universal kit (Qiagen), and RRBS libraries were constructed following standard protocol of the NuGEN Ovation Ultralow Methyl-seq Library Systems (TECAN)[58]. In all, 100 ng gDNA was used for library construction. The final libraries were quantified using Qubit 3.0 (Life Technologies), and the average size was determined using a TapeStation 2200 (Agilent Technologies). All libraries were sequenced on an Illumina HiSeq 4000 (Illumina, San Diego) with 100 bp single-end sequencing. Detailed sequencing information is listed in the Supplementary Table 3.

**RNA-seq gene expression analysis**. Briefly, the RNA-seq raw reads were trimmed using TrimGalore (v0.4.5) and then aligned to the human reference genome (GRCh38) for gene counts using STAR v2.5.4 (–quantMode Gene-Counts). RNA-seq raw fastq files from human primary bone-marrow derived

MSCs and adipose-derived MSCs were downloaded from GEO (SRR13081940, SRR13081941, SRR13081942, SRR17883873, SRR17883874, and SRR17883875). The data were processed using the same pipeline as the RNA-seq data generated in this study. DEGs were identified by DESeq2 (v1.26)[59] with FDR < 0.05 and fold change (FC) > 2. To perform principal component analysis (PCA), batch correction was performed using limma::removeBatchEffect(), and PCA was conducted using ggplot2 (v3.3.2). Heatmaps were generated using normalized read counts after log2() transformation and plotted using the pheatmap package (v1.0.12). Volcano plots were computed using the EnhancedVolcano (v1.4) package.

**ATAC-seq bioinformatic analysis and peak calling**. ATAC-seq raw reads were mapped to the human reference genome (GRCh38) using the bioinformatics pipeline snakePipes[60] in ATAC-seq mode. In the pipeline, fastq files were trimmed using TrimGalore and aligned with Bowtie2, and peaks were called with MACS2. A consensus peak set was defined by taking the intersection of peaks from both biological replicates using the soGGi (v1.18) package, and regions intersecting with blacklisted regions and ChrY were excluded. The ATAC-seq count matrix was computed by counting the reads that fell into the consensus peaks using Rsubread v2.0.1. Differential chromatin accessibility was generated by analyzing the normalized ATAC-seq count matrix using DESeq2 v1.26 for differential read counts. Peaks were annotated with genomic features using annotatr 1.21.1. Heatmaps were created using rlog-normalized pseudo-counts from the ATAC-seq count matrix using the pheatmap package (v1.0.12). Tracks were visualized with IGV v2.8.

**RRBS data analysis**. The RRBS raw fastq reads were trimmed using TrimGalore (v0.4.5) and Nugen RRBS trimming script (trimRRBSdiversityAdaptCustomers.py). After trimming, reads were aligned to the human reference genome (GRCh38) with Bismark[61] (v0.16.34) by default parameter settings, and PCR duplicates were removed with the Nugen script (nudup.py). The methylation call files (Bismark files), including the location of each CpG site and the methylation percentage, were generated by the bismark_methylation_extractor function. The methylation percentage per base from sorted Bismark files was determined using the processBismarkAln function from the methylKit package (v1.12). RRBS data files were processed with the methRead function, and the differentially methylated CpG (DMC) was generated by the getMethylDiff function (-difference = 20, $q$-value = 0.1) by methylKit (v1.12). These values were combined using the unite() function, and finally, only the regions that were covered by at least 10 reads were retained for further analysis. Heatmaps were generated using the beta values from DMCs with the pheatmap package (v1.0.12).

**Statistics and reproducibility**. Statistical analysis was performed using GraphPad Prism 8 or R version 3.6.3. Data are presented as the mean ± standard deviation (SD). Data comparison between two groups was performed using Student's $t$ test. Data comparisons between more than two groups were carried out using one-way ANOVA followed by Tukey's multiple comparisons test. A $P$ value of ≤0.05 was considered statistically significant. At least three biologically independent experiments for cell culture and flow cytometry experiments were performed.

**Reporting summary**. Further information on research design is available in the Nature Portfolio Reporting Summary linked to this article.

## Data availability

All the sequencing data have been uploaded to the NCBI GEO (Gene Expression Omnibus) under accession number GSE193201. Source data for the graphs are provided as Supplementary Data 1–10. Any additional information required is available from the corresponding authors.

## Code availability

We used many publicly available algorithms, code and packages for the RRBS, ATAC-seq, and RNA-seq mapping, genome annotation, differential analysis, etc., which were cited properly in the manuscript. This paper did not produce original code.

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

## Acknowledgements

This work was funded in part by American Heart Association (AHA) grant 18IPA34170301 (Charles W.), National Institutes of Health (NIH) grant S10OD019960 (Charles W.), Ardmore Institute of Health grant 2150141 (Charles W.), and Dr. Charles A. Sims' gift to LLU Center for Genomics. This study was also partially supported by NIH grants HL115195 (H.Q.)/subcontract (Charles W.), and HL137962 (H.Q.)/sub-contract (Charles W.). The authors are also appreciative to the partial funds provided by the LLU GCAT (Charles W. and M.B.).

## Author contributions

Charles W., X.B.Z., and H.Q. conceptualized and designed the study, supervised the work, and revised the manuscript; W.C., Chenguang W., Z.X.Y., F.Z., and W.W. carried out the experiments and collected the data; W.C. interpreted the data and drafted the manuscript; C.S., H.Q., and X.M. provided scientific input, revised the manuscript; Charles W. and M.B. acquired funding and provided resources.

## Competing interests

The authors declare no competing interests.
