## [Peer Review File · Communications Biology]

Reviewers' comments:

Reviewer #1 (Remarks to the Author):

In the present study entitled as 'Reprogramming of human peripheral blood mononuclear cells into induced mesenchymal stromal cells', Chen et al demonstrated that induced MSCs (iMSCs) can be generated by direct reprogramming of human peripheral blood mononuclear cells (PBMCs) using 5 factor transfection, including OCT4, SOX9, MYC, KLF4, and BCL-XL. The authors also suggested that OCT4 might serve as an universal reprogramming factor by epigenetic regulation by showing the impairment in MSC functionality in the absence of OCT4 transfection. I find this study to be interesting and novel in relevant field and helpful for the researchers in MSC research field. The study design is appropriate to prove authors' hypothesis and the results are solid. Overall, the manuscript is well-written. The manuscript will be more interesting and helpful for readers if the authors could add some experimental results showing the improved functions of iMSCs as therapeutics, such as the passaging limitation of iMSCs and crucial paracrine factors for from iMSCs, compared to normal MSCs isolated from tissues. Once this suggestion is appropriately reflected in the manuscript, the article may be appropriate for its publication.

Reviewer #2 (Remarks to the Author):

This study of Chen et al entitled as 'reprogramming of human peripheral blood mononuclear cells into induced mesenchymal stromal cells' aims to directly generate induce MSC (iMSCs) from PBMCs with the help of five transcription factors OCT4, SOX9, MYC, KLF4 and BCL-XL. The title says that the authors have used PBMCs and the experiments are done in CD34+ cells for reprogramming. Moreover, the authors identified that OCT4 is essential for the reprogramming of PBMCs into iMSCs. Even though, the authors put together major efforts, some of the data are not strong enough to support the conclusion. To improve the quality of the manuscript, authors need to address some of the major comments given below.

Comments:

1. The same group have already published in 'Cell Research' that rapid and efficient way of reprogramming blood CD34+ cells into iMSCs with single factor. In this manuscript, authors have used additional factors to reprogram CD34+ cells but not PBMCs. It will be good if the authors compare their own work single vs. 5 factors to show the advantages.
2. How do you know that your blood cells are not mixed from the blood MSCs. Are you purifying the CD34+ cells before reprogramming?
3. In Fig 1d and 2c, FACS data require proper controls (isotype and negative cell controls).
4. In Fig 1d, why you use only the TRA1-60 why not the other important pluripotent markers OCT4, Nanog and Sox2?
5. What about the important MSC marker CD105? Are these iMSCs maintaining their mesenchymal properties for longer period of time ie at higher passages?
6. It is essential to know when the OCT4 appears and disappears in reprogrammed iMSCs. For this, authors plan some experiments such as qRT-PCR or FACS analysis on cells collected at different time points.
7. How do the authors rule out the teratoma formation by the reprogrammed iMSCs generated from the CD34+ cells?
8. It is important to know that the direct reprogramming and prolonged culture may cause any chromosomal abnormalities. This can be analyzed by karyotyping or SNP analyses.
9. The functional data on iMSCs are missing.
10. All your FACS data need to be supported by additional experiments such as qRT-PCR or Western blot analysis. The single FACS data is not sufficient to draw the final conclusions.

Our **point-by-point responses** to Reviewers' comments are as following:

Reviewer #1 (Remarks to the Author):

In the present study entitled as 'Reprogramming of human peripheral blood mononuclear cells into induced mesenchymal stromal cells', Chen et al demonstrated that induced MSCs (iMSCs) can be generated by direct reprogramming of human peripheral blood mononuclear cells (PBMCs) using 5 factor transfection, including OCT4, SOX9, MYC, KLF4, and BCL-XL. The authors also suggested that OCT4 might serve as an universal reprogramming factor by epigenetic regulation by showing the impairment in MSC functionality in the absence of OCT4 transfection. I find this study to be interesting and novel in relevant field and helpful for the researchers in MSC research field. The study design is appropriate to prove authors' hypothesis and the results are solid. Overall, the manuscript is well-written. The manuscript will be more interesting and helpful for readers if the authors could add some experimental results showing the improved functions of iMSCs as therapeutics, such as the passaging limitation of iMSCs and crucial paracrine factors for from iMSCs, compared to normal MSCs isolated from tissues. Once this suggestion is appropriately reflected in the manuscript, the article may be appropriate for its publication.

Response: We would like to thank the reviewer for this comment and the great suggestion. We agree with the reviewer that the functional data of iMSCs is important. We have done additional experiments, now assessed the immunomodulatory properties of the 5F iMSCs. Briefly, we cultured PBMCs without or in the presence of 5F iMSCs (co-culture). T cells were stimulated with "Dynabeads™ Human T-Activator CD3/CD28" (ThermoFisher Scientific), and CD4 positive and CD8 positive T cells were assessed by flow cytometry on day 3 and day 6 of (co-)culture. We found that our 5F iMSCs were able to significantly suppress T-cell proliferation (CD4⁺ and CD8⁺ T-cell subsets) after both 3 days and 6 days of co-culture with PBMCs (**Fig. R1** below. **Fig. 2f** and **Suppl. Figure 4a** in the revised manuscript).

Figure R1. 5F iMSCs significantly inhibited T-cell proliferation after 3-day and 6-day co-culture with PBMCs.

We also evaluated crucial paracrine factors expressed by iMSCs reprogrammed with 5F, 4FnoO, or 4FnoK, as the reviewer suggested, and compared with the RNA-seq data of human bone marrow-derived MSCs (BMMSC)¹. We compared a list of major immunoregulatory cytokines, chemokines, and soluble factors secreted by MSCs^{2, 3} using the normalized gene counts from the RNA-seq data. This comparison is shown as bar graphs in a new **Suppl. Fig. 4b**. We found that when reprogramming

without OCT4, in addition to impaired tri-lineage differentiation potential, the 4FnoO iMSCs showed significantly reduced expression of many immunoregulatory cytokines/chemokines genes, such as IL-6, IL-10, HGF, VCAM1, CCL2, CXCL14 (**Suppl. Fig. 4b**). Both 5F and 4FnoK iMSCs generated in our current study showed comparable level of immunoregulatory cytokines/chemokines gene expression compared to the bone marrow derived MSCs.

All the data have been included in the revised manuscript result section, **Figure 2f**, and **Suppl. Figure 4**. We made some changes (highlighted) in the result section as follows:

On Page 6:

“Next, we evaluated the immunomodulatory potential of the iMSCs. We found that our 5F iMSCs were able to significantly suppress T-cell proliferation (CD4⁺ and CD8⁺ T cell subsets) after 3 (**Fig. 2f**) or 6 days co-culture with PBMCs (**Suppl. Fig. 4a**).”

On Page 8-9:

“To evaluate the immunomodulatory potentials of iMSCs reprogrammed with 5F, 4FnoO, or 4FnoK, we compared a list of major immunoregulatory cytokines, chemokines, and soluble factors secreted by MSCs^{2, 3} using the normalized gene counts from the RNA-seq data (**Suppl. Fig. 4b**). We found that compared with 5F iMSCs, in addition to impaired tri-lineage differentiation potential, the 4FnoO iMSCs showed significantly reduced gene expression on many immunoregulatory cytokines/chemokines, such as IL-10, HGF, VCAM1, CCL2, CXCL14 (**Suppl. Fig. 4b**). Both 5F and 4FnoK iMSCs showed comparable level of immunoregulatory cytokines/chemokines gene expression compared to the primary human bone marrow-derived MSCs¹.”

Supplementary Figure 4. a. iMSCs significantly inhibited T-cell proliferation after 6 days co-culture with PBMCs. **b.** The bar plotting showing RNA-seq gene expression values of the representative genes of immunomodulatory cytokines, chemokines, and soluble factors secreted by MSCs. Human primary MSCs, i.e., bone marrow-derived MSCs (BMMSC)¹ derived RNA-seq gene expression data was used as control. RNA-seq gene expression levels are shown as log₂() normalized read counts. n=3 in each group. * $P < 0.05$ indicates the statistically significance using BMMSC as control group; # $P < 0.05$ indicates the statistically significance using the 5F MSC as control group; error bars indicate standard deviation.

In addition, as suggested by the Reviewer 2, we have also compared the iMSCs from our current study with the iMSCs generated using our previously published method⁴, i.e., the Sca1-OCT4 iMSCs from our previous report⁴. We extracted RNA from the Sca1-OCT4 iMSCs and performed RNA-seq (three replicates) and compared a list of MSC-secreted immunoregulatory cytokines/chemokines^{2, 3} using the normalized gene counts from the RNA-seq data. We noticed that the Sca1-OCT4 iMSCs had a comparable gene expression level on certain important cytokines such as TGFB1, COX2 (**Fig. R2**), compared to the 5F iMSCs in our current study. We also found that the Sca1-OCT4 iMSCs showed a significant lower gene expression on several growth factors/chemokines, e.g., ANGPT1, VCAM1, CXCR3 and CXCL14, compared to the 5F iMSCs (**Fig. R2**). However, since this is not the focus of our current manuscript, we will not include these results in our current manuscript.

Figure R2. The bar plotting showing the RNA-seq gene expression values for the representative genes of immunomodulatory cytokines, chemokines, and soluble factors secreted by MSCs. RNA-seq gene expression levels are shown as $\log_2()$ normalized read counts. $n=3$ in each group. * $P < 0.05$, ** $P < 0.001$; all comparisons using the 5F iMSCs as control group; error bars indicate standard deviation.

Reviewer #2 (Remarks to the Author):

This study of Chen et al entitled as ‘reprogramming of human peripheral blood mononuclear cells into induced mesenchymal stromal cells’ aims to directly generate induce MSC (iMSCs) from PBMCs with the help of five transcription factors OCT4, SOX9, MYC, KLF4 and BCL-XL. The title says that the authors have used PBMCs and the experiments are done in CD34+ cells for reprogramming. Moreover, the authors identified that OCT4 is essential for the reprogramming of PBMCs into iMSCs. Even though, the authors put together major efforts, some of the data are not strong enough to support the conclusion. To improve the quality of the manuscript, authors need to address some of the major comments given below.

Comments:

1. The same group have already published in ‘Cell Research’ that rapid and efficient way of reprogramming blood CD34+ cells into iMSCs with single factor. In this manuscript, authors have used additional factors to reprogram CD34+ cells but not PBMCs. It will be good if the authors compare their own work single vs. 5 factors to show the advantages.

Response: We thank the reviewer for this suggestion. There are big differences between the two studies. The major differences between the current study with our previous work⁴ are: **1)** The cell source. In our previous work⁴, most of the results were generated using purified CD34+ cells derived from fetal cord blood. In the current study, we used adult PBMCs without purifying CD34+ cells. The reason for choosing adult peripheral blood cells over fetal cord blood CD34+ cells as source cells was due to their easy accessibility and availability, and, hence, has better clinical applicability and relevance. We aim to establish a simple system to easily generate large amounts of clinically relevant, integration-free

iMSCs from a few milliliters of peripheral blood. However, direct reprogramming of adult PBMCs has significantly reduced reprogramming efficiency compared to reprogramming CD34⁺ cord blood cells. Fetal CD34⁺ cells might be more versatile, and superior compared to adult CD34⁺ cells, not to mention that adult PBMCs only contain a very low percentage of CD34⁺ cells. This may explain why we were able to reprogram CD34⁺ cells enriched from fetal cord blood with one single factor in our previous study⁴, whereas we failed to direct reprogram adult PBMCs using the single factor OCT4 in this manuscript (**Fig. 1b**); **2**) The reprogramming method. In our previous work⁴, as a proof-of-concept, a significant portion of the results was generated using lentiviral vector-mediated OCT4 transduction. Compared to integration-free vectors, lentivirus transduction is easier to accomplish and achieves a high level of transgene expression. In the current study, all experiments were performed using an integration-free episomal vector system to directly reprogram adult PBMCs and generate integration-free iMSCs.

To better differentiate the current study from our previous work, we revised the title of this manuscript as follows:

“Reprogramming of human peripheral blood mononuclear cells into induced mesenchymal stromal cells using non-integrating vectors”.

We also made some changes (highlighted) in the Introduction section as follows:

On Page 3:

“Previously, we reported that fetal cord blood CD34⁺ cells can be directly reprogrammed into induced MSCs (iMSCs) by lentiviral delivery of OCT4 alone⁴. In this study, we choose adult peripheral blood mononuclear cells (PBMCs) as source cells due to their easy accessibility and availability compared to fetal cord blood. We aim to establish a simple system to easily generate large amounts of clinically relevant, integration-free iMSCs from a few milliliters of peripheral blood. In the present study, we found that unlike fetal CD34⁺ cells, a single factor OCT4 failed to reprogram adult PBMCs into iMSCs; however, through transient overexpression of five factors by episomal vectors (OCT4, BCL-XL, MYC, KLF4, and SOX9), adult PBMCs can be highly efficiently reprogrammed into integration-free iMSCs with trilineage differentiation potential. ”

2. How do you know that your blood cells are not mixed from the blood MSCs. Are you purifying the CD34⁺ cells before reprogramming?

Response: In this study, we used adult PBMCs for direct reprogramming without purifying CD34⁺ cells. In our opinion, the iMSCs reprogrammed from PBMCs in our study are not derived from the blood MSCs, because: **1**) The number of MSCs in peripheral blood are very scarce; **2**) Our prime culture conditions (6 days before reprogramming) support expansion of hematopoietic cells, but not MSCs; **3**) We did not observe MSC-like cells after 6 days of culture; **4**) When reprogramming with OCT4 only, no MSCs were observed. If there are preexisting MSCs, we should expect to observe MSC colonies after reprogramming with any factors or even with no factors but that was not the case; **5**) Most importantly, when we depleted the CD34⁺ in the PBMCs before reprogramming with five factors, non-MSC-like colonies were observed (**Fig. 1e**), suggesting the CD34⁺ cells in PBMCs are responsible for the generation of iMSCs and not the blood MSCs.

3. In Fig 1d and 2c, FACS data require proper controls (isotype and negative cell controls).

Response: As the reviewer suggested, we have added the proper controls in Fig. 1d and 2c in the revised manuscript, as shown below.

Figure 1. (d) Fluorescence-activated cell sorting (FACS) analysis of iMSCs 8 days after reprogramming with different factor combinations. SOX2 induced iPSCs generation (TRA-1-60⁺ cells). However, SOX9 did not induce detectable TRA-1-60⁺ cells.

Figure 2. (c) Flow cytometry plots of typical MSC marker expression (CD29, CD73, CD90, CD166)

4. In Fig 1d, why you use only the TRA1-60 why not the other important pluripotent markers OCT4, Nanog and Sox2?

Response: Thank you for your great suggestion. Since OCT4 and SOX2 were used as reprogramming factors in all or part of the groups, although these factors were delivered by non-integrating vectors, at 1-2 weeks after nucleofection, they might still be detectable as exogenous expression and will interfere with data interpretation. We have now performed FACS analysis for iMSCs using the pluripotent marker NANOG, which was not part of our reprogramming combination as the reviewer suggested.

In agreement with our TRA-1-60 data (Fig. 1d), we found that in the presence of SOX2, ~1-2% reprogrammed cells were NANOG⁺ (Suppl. Fig. 1). On the other hand, SOX9 did not induce detectable NANOG⁺ cells (Suppl. Fig. 1). These data have been included in the revised manuscript result section and Supplementary Figure 1 as follows:

On Page 4:

“However, the presence of SOX2 in the reprogramming cocktail resulted in ~1-2% of reprogrammed cells expressing iPSC markers, e.g., TRA-1-60 (**Fig. 1d**) and NANOG (**Suppl. Fig. 1**), even in MSC expansion culture conditions.”

Supplementary Figure 1. Fluorescence activated cell sorting (FACS) analysis of iMSCs showed that with the presence of SOX2 factor, ~ 1-2% reprogrammed cells were NANOG positive. Replace SOX2 with SOX9 did not induce detectable NANOG⁺ cells.

5. What about the important MSC marker CD105? Are these iMSCs maintaining their mesenchymal properties for longer period of time ie at higher passages? Did you look at this?

Response: Yes, we agree with the reviewer that CD105 is an important surface marker of mesenchymal stromal cells^{5, 6}, and we did include it as one of the ten well-established MSC surface markers when comparing their expression between primary MSCs and our iMSCs (**Suppl. Fig 11**). To better visualize the comparison, we plotted the log₂() transformation CD105 gene count from the RNA-seq data as a bar graph (**Fig. R3** below). As we mentioned in the manuscript, the expression of CD105 is comparable between both functional iMSCs (5F and 4FnoK iMSCs) and malfunctional iMSCs (4FnoO iMSCs).

Although it has been reported that CD105⁺ synovial membrane MSCs have stronger chondrogenic potential than CD105⁻ synovial membrane MSCs⁷, it was also noted that the CD105⁺ MSCs and CD105⁻ MSCs have similar stemness and *in vitro* tri-lineage differentiation potential^{8, 9}. Moreover, it has been shown that the expression of CD105 on MSCs was affected by the serum level present in the culture medium¹⁰. We believe that our results are in line with these data, suggesting that some MSC marker genes, such as CD105, might not truly correlate with the functionality of iMSCs.

Regarding the question about the culture duration, we have continuously cultured these iMSCs for ~2 months, and we did not observe any difference in phenotypes.

Figure R3. The bar graph shows the expression of the MSC surface marker CD105 on primary MSCs and iMSCs. AdMSC, Adipose-derived MSCs; BM, bone marrow-derived MSCs.

6. It is essential to know when the OCT4 appears and disappears in reprogrammed iMSCs. For this, authors plan some experiments such as qRT-PCR or FACS analysis on cells collected at different time points.

Response: In our previous study⁴, we have conducted real-time RT-PCR analysis to determine OCT4 expression in the reprogrammed cells. We found that when reprogramming CD34⁺ cells (enriched from fetal cord-blood) with episomal vector EV SFFV-OCT4, at one month after nucleofection, OCT4 expression in the iMSCs was barely detectable, decreased to baseline level comparable to bone marrow MSCs (**Suppl. Fig. 5** in our previous publication⁴). We further looked at the OCT4 gene count number in the RNA-seq data of the current study. We did not observe OCT4 expression in our iMSCs, suggesting no exogenous expression and no endogenous activation of OCT4 in the iMSCs derived in our current study.

Furthermore, we performed FACS analysis of OCT4 expression on iMSCs reprogrammed from PBMCs at one month after nucleofection and did not observe any OCT4⁺ cells (**Suppl. Fig. 3**). These data have been included in the revised manuscript Results' section and in **Supplementary Figure 3**.

Supplementary Figure 3. Fluorescence activated cell sorting (FACS) analysis of iMSCs at one month after nucleofection showed that no detectable OCT4⁺ cells.

7. How do the authors rule out the teratoma formation by the reprogrammed iMSCs generated from the CD34+ cells?

Response: In our previous study⁴, we evaluated the potential risk of tumor formation of the iMSCs reprogrammed from fetal cord blood CD34⁺ cells. We did not observe teratoma formation during 3 months of follow-up. In addition, after reprogramming, both our FACS data and gene count number from RNA-seq data showed no expression of typical iPSC markers, such as OCT4, TRA-1-60, and NANOG, in our reprogrammed iMSCs. Hence, it would be extremely unlikely that the iMSCs generated in the current study could form teratomas, and for this reason we did not perform such an experiment.

8. It is important to know that the direct reprogramming and prolonged culture may cause any chromosomal abnormalities. This can be analyzed by karyotyping or SNP analyses.

Response: We thank the reviewer for this comment. We have now assessed the karyotype of primary PBMCs and 5F iMSC cells cultured for up to one month. Briefly, genomic DNA were extracted from primary PBMCs and iMSCs cultured for one week (passage 3, P3) and four weeks (passage 10, P10), and DNA were hybridized to Infinium BeadChip (Illumina) followed by staining and scanning on the Illumina HiScan system. These arrays can interrogate 597,784 human SNP markers, thus yielding up to 50-fold better resolution (~100 kb) than conventional karyotyping by Giemsa banding. B allele frequency (BAF) and Log R ratio (LRR) were used to detect copy number variants (CNVs). When viewing the data, the blue points represent BAF, which is the proportion of hybridized sample that carries the B allele. The BAF values of 0.0, 0.5, and 1.0 for each locus (representing AA, AB, and BB) can be seen in a normal sample. The red points represent Log R ratio. Any deviations from 0.0 indicate copy number changes. In summary, after digital karyotyping by SNP arrays, we did not find any chromosomal abnormalities in both PBMCs and cultured 5F iMSCs. These data have been included in the revised manuscript and **Supplementary Figure 5-7**. We made changes (highlighted) in the Results section as follows:

On Page 6:

“To further determine if the reprogramming to iMSCs or their expansion in culture may cause any chromosomal abnormalities, we performed digital karyotyping using SNP arrays. We did not identify any chromosomal abnormalities after either one week or four weeks of *in vitro* culture (**Suppl. Fig. 5-7**).”

9. The functional data on iMSCs are missing.

Response: We agree with the reviewer that the functional data of iMSCs is important. In addition to the data showing the trilineage differentiation capability of iMSCs in this manuscript, we have done additional experiments and now assessed the immunomodulatory properties of the 5F iMSCs. Briefly, we cultured PBMCs with or without the co-culture of 5F iMSCs. T cells were stimulated with “Dynabeads™ Human T-Activator CD3/CD28” (ThermoFisher Scientific), and CD4 positive and CD8 positive T cells were assessed by flow cytometry on day 3 and day 6 of (co-)culture. We found that our 5F iMSCs were able to significantly suppress T-cell proliferation (CD4⁺ and CD8⁺ T-cell subsets) after both 3 days and 6 days of co-culture with PBMCs (**Fig. R1** below. **Fig. 2f** and **Suppl. Figure 4a** in the revised manuscript).

Figure R1. 5F iMSCs significantly inhibited T-cell proliferation after 3-day and 6-day co-culture with PBMCs.

We also evaluated crucial paracrine factors expressed by iMSCs reprogrammed with 5F, 4FnoO, or 4FnoK, as the reviewer suggested, and compared with the RNA-seq data of human bone marrow-derived MSCs (BMMSC)¹. We compared a list of major immunoregulatory cytokines, chemokines, and soluble factors secreted by MSCs^{2, 3} using the normalized gene counts from the RNA-seq data. This comparison is shown as bar graphs in a new **Suppl. Fig. 4b**. We found that when reprogramming without OCT4, in addition to impaired tri-lineage differentiation potential, the 4FnoO iMSCs showed significantly reduced expression of many immunoregulatory cytokines/chemokines genes, such as IL-6, IL-10, HGF, VCAM1, CCL2, CXCL14 (**Suppl. Fig. 4b**). Both 5F and 4FnoK iMSCs generated in our current study showed comparable level of immunoregulatory cytokines/chemokines gene expression compared to the bone marrow derived MSCs.

All the data have been included in the revised manuscript result section, **Figure 2f**, and **Suppl. Figure 4**. We made some changes (highlighted) in the result section as follows:

On Page 6:

“Next, we evaluated the immunomodulatory potential of the iMSCs. We found that our 5F iMSCs were able to significantly suppress T-cell proliferation (CD4⁺ and CD8⁺ T cell subsets) after 3 (**Fig. 2f**) or 6 days co-culture with PBMCs (**Suppl. Fig. 4a**).”

On Page 8-9:

“To evaluate the immunomodulatory potentials of iMSCs reprogrammed with 5F, 4FnoO, or 4FnoK, we compared a list of major immunoregulatory cytokines, chemokines, and soluble factors secreted by MSCs^{2, 3} using the normalized gene counts from the RNA-seq data (**Suppl. Fig. 4b**). We found that compared with 5F iMSCs, in addition to impaired tri-lineage differentiation potential, the 4FnoO iMSCs showed significantly reduced gene expression on many immunoregulatory cytokines/chemokines, such as IL-10, HGF, VCAM1, CCL2, CXCL14 (**Suppl. Fig. 4b**). Both 5F and 4FnoK iMSCs showed comparable level of immunoregulatory cytokines/chemokines gene expression compared to the primary human bone marrow-derived MSCs¹.”

Supplementary Figure 4. a. iMSCs significantly inhibited T-cell proliferation after 6 days co-culture with PBMCs. **b.** The bar plotting showing RNA-seq gene expression values of the representative genes of immunomodulatory cytokines, chemokines, and soluble factors secreted by MSCs. Human primary MSCs, i.e., bone marrow-derived MSCs (BMMSC)¹ derived RNA-seq gene expression data was used as control. RNA-seq gene expression levels are shown as log₂(normalized read counts). n=3 in each group. * $P < 0.05$ indicates the statistically significance using BMMSC as control group; # $P < 0.05$ indicates the statistically significance using the 5F MSC as control group; error bars indicate standard deviation.

10. All your FACS data need to be supported by additional experiments such as qRT-PCR or Western blot analysis. The single FACS data is not sufficient to draw the final conclusions.

Response: We appreciate the reviewer raising this point. However, we would like to emphasize that FACS data shows protein expression at the single cell level, whereas Western blot analysis only shows protein expression in bulk populations, and qRT-PCR looks at transcript levels which may not always correlate with protein expression levels (discordance between RNA and protein expression). In addition, the conclusions of our current study are not only supported by FACS data, but also by many other data generated using different types of experiments/approaches, including RNA-seq, ATAC-seq, RRBS, colony formation, *in vitro* tri-lineage differentiation etc. Hence, we are of the opinion that our drawn conclusions are based on a variety of solid data.

References cited:

1. Rubinstein-Achiasaf L, Morein D, Ben-Yaakov H, Liubomirski Y, Meshel T, Elbaz E, et al. Persistent inflammatory stimulation drives the conversion of mscs to inflammatory cacs that promote pro-metastatic characteristics in breast cancer cells. *Cancers (Basel)*. 2021;13
2. Kyurkchiev D, Bochev I, Ivanova-Todorova E, Mourdjeva M, Oreshkova T, Belemezova K, et al. Secretion of immunoregulatory cytokines by mesenchymal stem cells. *World J Stem Cells*. 2014;6:552-570
3. Song N, Scholtemeijer M, Shah K. Mesenchymal stem cell immunomodulation: Mechanisms and therapeutic potential. *Trends in Pharmacological Sciences*. 2020;41:653-664
4. Meng X, Su RJ, Baylink DJ, Neises A, Kiroyan JB, Lee WY, et al. Rapid and efficient reprogramming of human fetal and adult blood cd34+ cells into mesenchymal stem cells with a single factor. *Cell Res*. 2013;23:658-672
5. Barry FP, Boynton RE, Haynesworth S, Murphy JM, Zaia J. The monoclonal antibody sh-2, raised against human mesenchymal stem cells, recognizes an epitope on endoglin (cd105). *Biochemical and biophysical research communications*. 1999;265:134-139
6. Consentius C, Mirenska A, Jurisch A, Reinke S, Scharm M, Zenclussen AC, et al. In situ detection of cd73+ cd90+ cd105+ lineage: Mesenchymal stromal cells in human placenta and bone marrow specimens by chipcytometry. *Cytometry Part A*. 2018;93:889-893
7. Chang C, Han S, Kim E, Lee S, Seong S, Lee M. Chondrogenic potentials of human synovium-derived cells sorted by specific surface markers. *Osteoarthritis and Cartilage*. 2013;21:190-199
8. Pham LH, Vu NB, Van Pham P. The subpopulation of cd105 negative mesenchymal stem cells show strong immunomodulation capacity compared to cd105 positive mesenchymal stem cells. *Biomedical Research and Therapy*. 2019;6:3131-3140
9. Cleary M, Narcisi R, Focke K, Van der Linden R, Brama P, van Osch G. Expression of cd105 on expanded mesenchymal stem cells does not predict their chondrogenic potential. *Osteoarthritis and cartilage*. 2016;24:868-872
10. Mark P, Kleinsorge M, Gaebel R, Lux CA, Toelk A, Pittermann E, et al. Human mesenchymal stem cells display reduced expression of cd105 after culture in serum-free medium. *Stem Cells International*. 2013;2013

REVIEWERS' COMMENTS:

Reviewer #1 (Remarks to the Author):

It seems that the authors resolved all of the issues that I raised in the last review. The authors have addressed the various points raised by conducting additional experiments and the manuscript is improved. I think the manuscript is appropriate for its acceptance and publication in its current form.

Reviewer #2 (Remarks to the Author):

The authors are responsive and answered all my comments. No more concerns on this manuscript.